# DIP/Dpr interactions and the evolutionary design of specificity in protein families

Alina P. Sergeeva [1], Phinikoula S. Katsamba[2], Filip Cosmanescu[2], Joshua J. Brewer[3], Goran Ahlsen[2], Seetha Mannepalli[2], Lawrence Shapiro[2,3] ✉ & Barry Honig[1,2,3,4] ✉

Differential binding affinities among closely related protein family members underlie many biological phenomena, including cell-cell recognition. *Drosophila* DIP and Dpr proteins mediate neuronal targeting in the fly through highly specific protein-protein interactions. We show here that DIPs/Dprs segregate into seven specificity subgroups defined by binding preferences between their DIP and Dpr members. We then describe a sequence-, structure- and energy-based computational approach, combined with experimental binding affinity measurements, to reveal how specificity is coded on the canonical DIP/Dpr interface. We show that binding specificity of DIP/Dpr subgroups is controlled by "negative constraints", which interfere with binding. To achieve specificity, each subgroup utilizes a different combination of negative constraints, which are broadly distributed and cover the majority of the protein-protein interface. We discuss the structural origins of negative constraints, and potential general implications for the evolutionary origins of binding specificity in multi-protein families.

[1] Department of Systems Biology, Columbia University, New York, NY, USA. [2] Zuckerman Mind, Brain and Behavior Institute, Columbia University, New York, NY, USA. [3] Department of Biochemistry and Molecular Biophysics, Columbia University, New York, NY, USA. [4] Department of Medicine, Columbia University, New York, NY, USA. ✉email: lawrenceshapiro@gmail.com; bh6@cumc.columbia.edu

Over the course of evolution, gene duplications followed by sequence divergence have generated numerous protein families whose homologous members have distinct binding specificities. In many cases there are only subtle differences in the sequence and structure of closely related family members, yet these can have profound functional consequences. Families of cell–cell adhesion proteins offer many such examples, where seemingly small changes in sequence generate precise protein–protein interaction specificities. For example, invertebrate Dscam[1,2] and vertebrate clustered protocadherin[3,4] neuronal "barcoding" proteins each display strict intra-family homophilic specificity. While one Dscam (or Pcdh) isoform may have >90% sequence identity to another family member, the few differences between them ensure that only homophilic recognition occurs. This is critical to their function in the diversification of neuronal identities and self/non-self-discrimination[1–4]. In other cases, for example the type I and type II classical cadherins which pattern epithelia and other tissue structures, specificity is less strict[5–7]. While classical cadherins also exhibit homophilic binding, in addition they show strong heterophilic binding to other select family members[5–7]. In another example, the nectin adhesion protein family encodes interactions between members that are mainly heterophilic, contributing to the formation of a checkerboard pattern between two cell types, each expressing a cognate nectin[8]. The specificity patterns of these protein families underlie their biological functions, and are conserved in evolution. Understanding how specificity is coded on multi-protein adhesion protein families is thus critical.

Here, we carry out a comprehensive computational and experimental study of specificity determinants in two interacting families of *Drosophila melanogaster* neuronal recognition proteins, the 21-member Dpr (Defective proboscis extension response) and the 11-member DIP (Dpr Interacting Proteins). These proteins have been extensively characterized structurally[9–11], and their interactions were characterized quantitatively with biophysical measurements[11]. They thus offer an ideal system to study the evolutionary design of specificity on protein–protein interfaces. DIPs and Dprs are expressed in cell-specific patterns throughout the developing nervous system[12]. DIPs preferentially bind Dprs, and a network of specific heterophilic interactions is formed between members of the two families. This molecular binding network is correlated with synaptic specificity in the fly retina, suggesting that DIP/Dpr interactions play an important role in neuronal patterning[9,13].

The extracellular regions of DIP and Dpr family members consist of three and two tandem Ig-like domains, respectively[13]. Homodimerization is observed for some DIPs and Dprs, and homo-dimerization and hetero-dimerization is mediated by an interface formed between the membrane-distal Ig1 domains (Fig. 1a). Surface plasmon resonance (SPR) showed that members of both families have distinct binding profiles, with DIPs and Dprs initially classified as forming four specificity subgroups[11]. In the current work we extended the number of subgroups to seven based primarily on the strongest heterophilic binding preferences but also on DIP/Dpr sequence similarity (see color-coded subgroup assignment in Fig. 1b). Our DIP/Dpr grouping is somewhat different than that published by Cheng et al.[10] due in part to the fact that these authors did not include DIP-κ and DIP-λ, whose binding preferences had been previously mapped[11]. Additional differences could be due to the biophysical approaches used to measure DIP/Dpr binding affinities in Cosmanescu et al.[11] and Cheng et al.[10] (see Methods section).

The Ig1 domains of the *Drosophila melanogaster* DIPs and Dprs have intra-family pairwise sequence identities greater than about 50% and 40%, respectively, while the average identity between individual DIPs and Dprs is about 30%. Binding interfaces for crystallographically determined hetero-dimer structures are essentially identical—superimposing to within 1 Å (ref.[11]) (Fig. 1c). The central question we address here is how DIPs and Dprs that are so closely related in sequence and structure can exhibit such highly specific pairwise interactions. Previous studies have identified specificity residues for select DIP/Dpr interactions[9–11]. Here, we analyze specificity for the family as a whole. Our results reveal the central role of "negative constraints", used here to denote an amino acid in a cognate interface that interferes with binding to a non-cognate partner. The term negative constraint has been used in the field of protein design[14–17] to denote a domain that must be designed against, in effect an "anti-target". By contrast, our use of the term here focuses on individual amino acids rather than entire domains.

Since there are a total of forty-nine possible DIP/Dpr subfamily pairs and only seven bind strongly, there must be forty-two sets of negative constraints that preclude incorrect pairing. These are coded on a pseudo-symmetric Ig1–Ig1 interface of about 1900 Å$^2$ buried surface area and comprising 33 interfacial residues on the DIP side and 33 interfacial residues on the Dpr side. We recognize that non-interfacial residues may also contribute to specificity but the primary determinants will, in most cases, be part of the interface and these are the focus of the current work.

We begin by asking what can be learned from sequence alone and find that this information is useful but incomplete. Our structure-based approach involves building homology models of hypothetical complexes formed between all DIPs and Dprs and, using solved complexes where available, to calculate the energetic consequences of mutating interfacial residues in one family member to those of every other. We tested various programs that calculate binding free energy changes ($\Delta\Delta G$s) resulting from mutations and found that FoldX[18] offered the best combination of accuracy and computational efficiency. Using FoldX, we identified negative constraints for every DIP/Dpr pair and then confirmed a subset of our predictions using SPR.

Our results provide a detailed account of the design of complex specificity on a canonical protein–protein interface. Most of the interface is used to create negative constraints with virtually all expected energetic terms (steric hindrance, Coulombic repulsion, unsatisfied buried charges, and/or polar groups) used to weaken undesired complex formation between different subgroup pairs. Some locations on the interface are used by multiple pairs while others are used in only a small number of cases. In general, more than one negative constraint is required to significantly weaken binding. We also discuss the trade-off between specificity and affinity, a phenomenon that appears essential for the generation of multiple specificity groups[14,15,19]. Overall, our analysis reveals general principles about the design of protein families whose members are very similar in sequence and structure and yet exhibit exquisitely controlled binding affinities and specificities.

## Results

**Sequence analysis of DIPs and Dprs.** Figure 2 plots an alignment of sequence logos corresponding to interfacial residues of DIP and Dpr *Insecta* orthologs (see Methods section for details). Interfacial positions of DIPs and Dprs were assigned based on non-zero change in accessible surface area per residue upon complex formation in experimentally solved crystal structures. Interfacial positions of DIP/Dpr complexes of unknown structure were inferred from multiple sequence alignment (residue correspondence with interfacial positions of solved complexes). The family-wide numbering of interfacial positions ($i = 1$–33, designated above the logos in Fig. 2) is used throughout the paper (see correspondence between the family-wide numbering and Uniprot

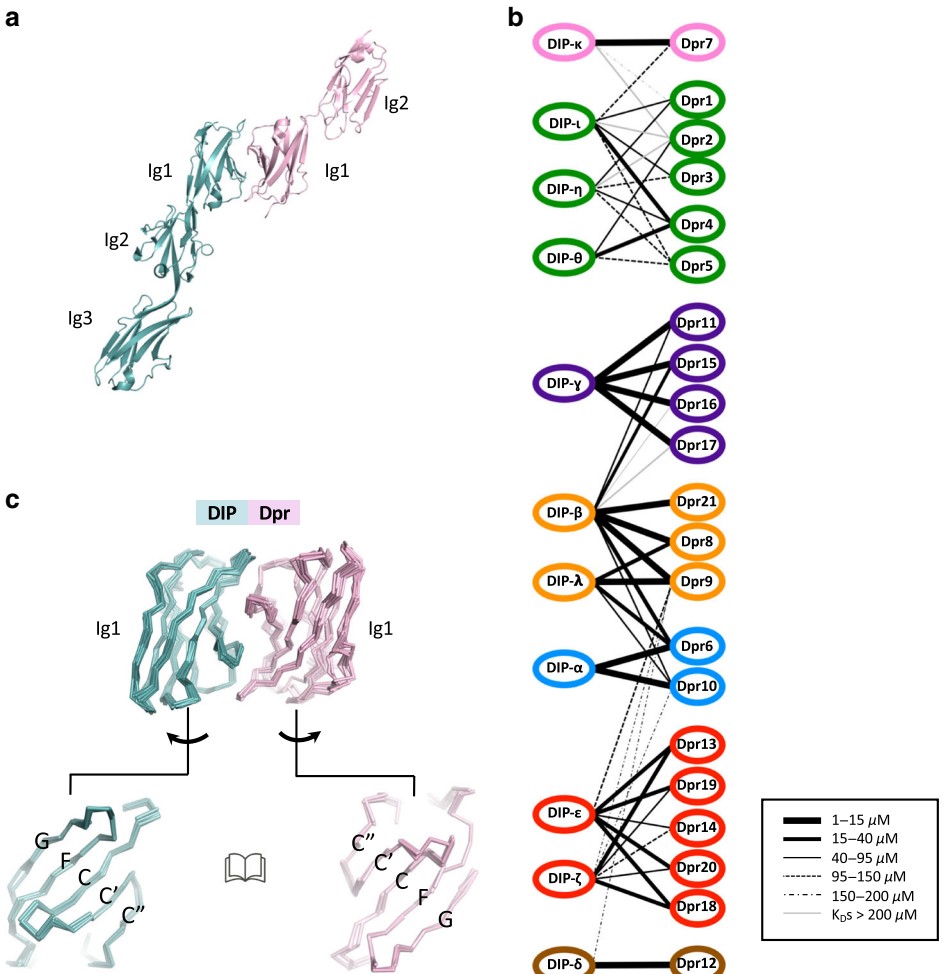

**Fig. 1 Structure and interaction properties of DIPs and Dprs. a** Ribbon representation of the DIP/Dpr heterodimer (PDBID: 6EG0)[11]—DIP shown in cyan, Dpr in pink. **b** Affinity-based binding interactome of DIPs and Dprs of *Drosophila melanogaster*. Line thickness indicates the $K_D$ for a given pairwise interaction, as specified in the boxed inset. Groupings are color-coded based on their heterophilic binding preference. The interactome is based on previously published SPR data[11]. **c** Structural alignment of Ig1 domains in DIP/Dpr complexes (see Methods section for PDBIDs). DIPs and Dprs interact via their CC'C"FG surfaces.

numbering (given throughout this paper in curly brackets) in Supplementary Table 1).

Four positions shown in yellow are occupied almost exclusively by hydrophobic residues which form the hydrophobic core of the DIP/Dpr interface[10] (Fig. 2a). Additionally, shown in purple is a conserved Gln in both subfamilies and a conserved Asn in DIPs, which interlock the Ig1–Ig1 interface by forming buried sidechain-to-backbone hydrogen bonds (Fig. 2b). Residues that are conserved across subfamilies cannot be responsible for subgroup specificity but changes in the identity of individual conserved hydrophobic residues can play a role (see below). Rather, potential specificity determinants will generally correspond to positions that are conserved in at least one subfamily but differ in between subfamilies.

We tested the ability of a number of sequence-based methods to identify specificity determinants in the seven specificity subgroups. These include GroupSim[20], SDPpred[21], SPEER[22], and Multi-Harmony[23]. As input, we used a multiple sequence alignment of just interfacial residues with protein sequences of 1732 DIP or 2570 Dpr orthologs segregated into seven specificity subgroups (see Fig. 2 and Methods section for details). Nine specificity determining positions were identified based on a consensus of at least 3 out of 4 of these methods, and 14

additional positions, were predicted by at least one method (see predictions below logos in gray, Fig. 2). These expand the number of predicted specificity determining sites compared to our previous study[11] where we relied on visual inspection of a multiple sequence alignment of DIP/Dpr sequences in *Drosophila melanogaster* alone. Below we compare the predictions of these sequence-based methods to an energy-based analysis of DIP and Dpr structures.

**Identifying negative constraints.** We hypothesized that negative constraints are the essential factor that defines DIP/Dpr subgroups and developed a computational strategy to identify and characterize these putative constraints. Since each subgroup (e.g., i and j) has both DIP and Dpr members, for purposes of discussion we can define four sets of proteins, DIP-i, DIP-j, Dpr-i, and Dpr-j. We need to explain why members of set DIP-i don't bind to members of set Dpr-j and, in turn, why members of set DIP-j don't bind to members of set Dpr-i. Since Dpr-i binds to DIP-i while Dpr-j does not, we first use FoldX calculations to identify residues of Dpr-i that weaken binding to DIP-i when mutated into a residue of Dpr-j. Thus, we identify negative constraints of Dpr-j by studying their predicted effects on the

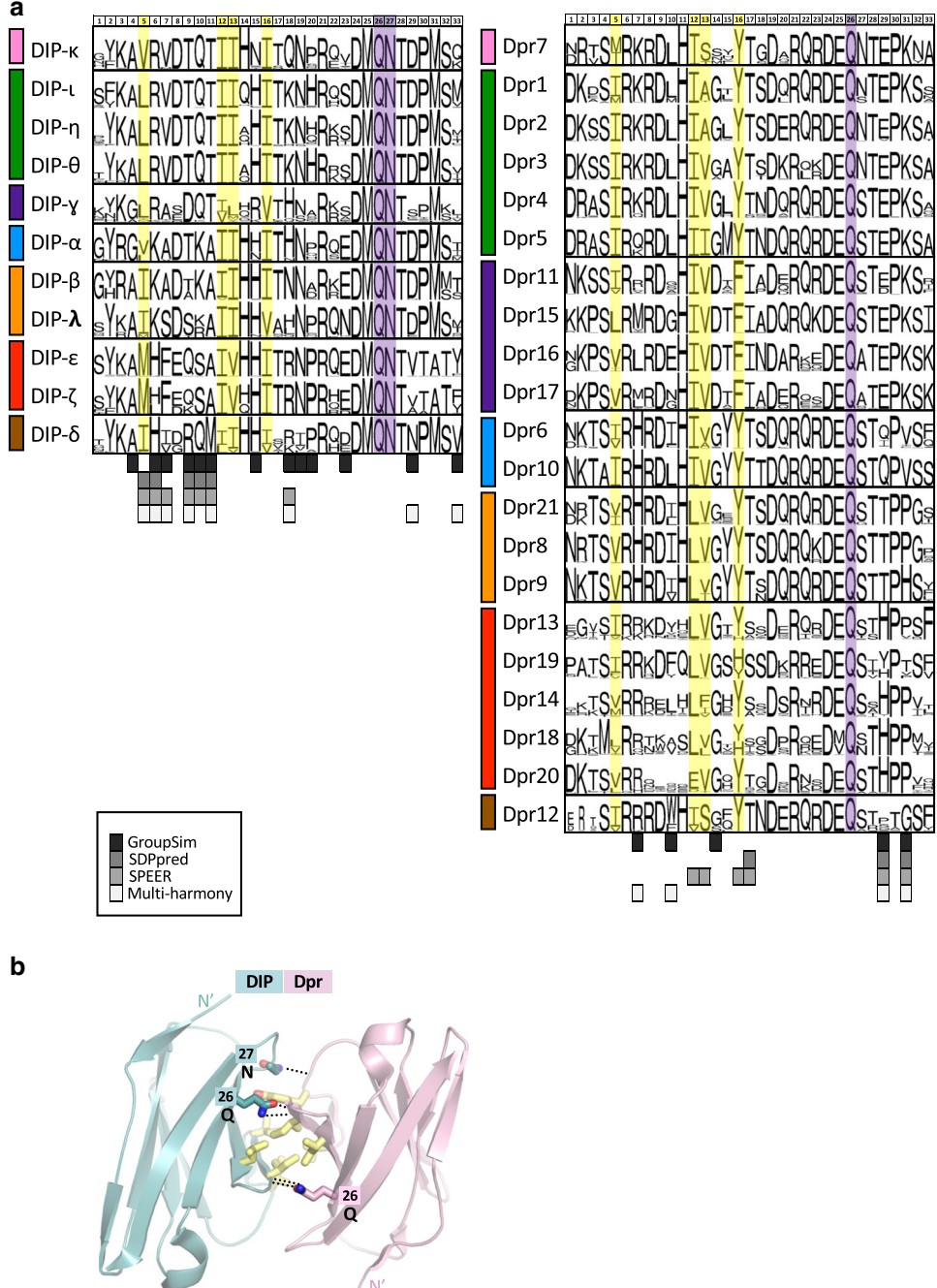

**Fig. 2 Sequence-based predictions of DIP/Dpr specificity residues and common binding determinants. a** Alignment of sequence logos corresponding to interfacial residues of DIP and Dpr *Insecta* orthologs. Predictions of specificity determining positions using sequence-based methods (GroupSim, SDPpred, SPEER, and Multi-Harmony) are given below logos in different shades of gray as specified in boxed inset. Bar on the left of logos indicates color of the DIP/ Dpr subgroup (Fig. 1b). Family-wide numbering of interfacial positions is given above the alignment. The correspondence between protein specific (UniProt) and family-wide numbering is given in Supplementary Table 1. See Methods section for details on sequence logo preparation. **b** Common binding determinants of DIP/Dpr dimers depicted on ribbon Ig1-Ig1 interfaces in sticks. Hydrophobic interfacial residues are in yellow. Conserved buried hydrogen bonds are depicted by dotted lines. Interfacial positions enclosed in boxes follow family-wide numbering and are annotated with identities of conserved amino-acids.

binding of mutated Dpr-i to DIP-i. The experimental tests follow the same logic: the $K_D$s of mutant Dpr-i binding to DIP-i are used to validate predicted negative constraints on Dpr-j.

The overall procedure was to start with one or more structures of DIP/Dpr complexes for each subgroup (the template complex) and (1) to mutate every interfacial residue in the DIP protomer to all residues appearing at the same position in DIPs of other

specificity subgroups and then (2) calculate ΔΔGs for the binding of each DIP to the Dpr in the complex. (3) Repeat the procedure for the Dpr in the complex and then calculate ΔΔGs for its binding to each DIP. Templates that were used include: blue subgroup—crystal structure of Dpr6/DIP-α and two conformations of Dpr10/DIP-α[9,10]; purple subgroup—crystal structure of Dpr11/DIP-γ[10]; green subgroup—crystal structures of Dpr4/DIP-

η and for two conformations each for Dpr1/DIP-η and Dpr2/DIP-θ)[10,11]; pink subgroup—homology model of Dpr7/DIP-κ; orange subgroup—homology model of Dpr8/DIP-β; brown subgroup—homology model of Dpr12/DIP-δ; red subgroup—homology model of Dpr13/DIP-ε. Overall, we predicted the effects 3044 mutations of 33 interfacial residues of 13 template complexes on the DIP and Dpr side into residues occurring in DIPs and Dprs of other subfamilies (Supplementary Data 1). Notably, the results depend heavily on the use of homology models (see Methods section for details of how they were constructed).

Negative constraints were defined as positions predicted to cause destabilizing effects for every DIP or Dpr fly subgroup member in the context of binding to a member of another Dpr or DIP subgroup, respectively. Our energy-based filter also requires that the effect is destabilizing for every subfamily template complex used in the FoldX calculations (see "energy filter" in Methods section for details). In addition, we introduced a second filtering step, whereby we only defined a negative constraint as a position that would also be predicted for other *Insecta* species so as to focus on negative constraints that are evolutionarily conserved. The evolutionary filter assumes that orthologs in the same subfamily conserve their interaction specificity (see Methods section for data supporting the assumption). To clarify, the energy-based filter is based on FoldX calculations of structures of fly proteins whereas the evolutionary filter is based on comparison of ortholog sequences of different species of insects. The evolutionary filter involved an analysis of interfacial residues at corresponding positions of DIP-i and DIP-j *(or Dpr-i and Dpr-j)* in *Insecta* multiple sequence alignments to see if the identities, size, or biophysical properties of amino acids in all members of subfamily "i" were different from those of subfamily "j" at the predicted position (see "evolutionary filter" in Methods section for details).

A subset of our predictions (see below) was tested experimentally. Overall, we identified negative constraints for 42 combinations of non-interacting DIP and Dpr subgroups (Supplementary Data 1).

**Evaluation of methods to calculate ΔΔG.** We tested six algorithms that calculate binding affinity changes upon mutation of individual residues (ΔΔGs) based on reports of their performance in the literature[24–30]. These include FoldX[18,31], mCSM[30], BeAtMusic[27], MutaBind[29], Rosetta flex ddG[24], and BindProfX[32]. FoldX, mCSM, and BeAtMusic require from seconds to few minutes of CPU time per single point mutation, while MutaBind, Rosetta flex ddG, and BindProfX take more than an hour. Our evaluation tests were based on 25 experimental ΔΔG values obtained via SPR or AUC measurements of DIP/Dpr and DIP/DIP dimers. Of these, two were published[11] and twenty-three are new (see Supplementary Fig. 1 for supporting AUC and SPR data). Data on the performance of these methods is summarized in Supplementary Table 2.

As shown in Supplementary Table 2, FoldX, MutaBind, and Rosetta flex ΔΔG perform best on our dataset as they feature the highest Pearson correlation coefficients (PCC). MutaBind is the best performer but it fails to identify stabilizing mutations (shown in green, Supplementary Table 2)—a failure of all machine learning methods we tested, which is probably reflective of an over-representation of destabilizing mutations in training datasets. In addition, we preferred to use methods based on physics-based force fields since these were easiest to interpret in structural terms. Given its performance in terms of speed and accuracy (PCC = 0.57, Supplementary Table 2), we settled on FoldX as a computational guide for the identification of negative constraints.

**Negative constraints between blue and purple subgroups.** We discuss Dpr10/DIP-α (blue subgroup) and Dpr11/DIP-γ (purple subgroup) in detail since these pairwise DIP/Dpr interactions have revealed clear associated phenotypes in the central and peripheral nervous systems of *Drosophila melanogaster*[10,33,34]. We consider four sets of negative constraints that weaken blue/purple inter-subgroup DIP/Dpr binding: (a) On purple subgroup Dprs; (b) On blue subgroup DIP-α; (c) On blue subgroup Dprs; and (d) On purple subgroup DIP-γ.

a. Using the protocol summarized above, we identify negative constraints on purple subgroup Dprs that weaken binding to blue subgroup DIP-α by computationally mutating every interfacial amino acid on blue Dprs to those of purple Dprs and calculating ΔΔGs for binding to DIP-α. The protocol is applied for every template complex of the blue subfamily listed above (Supplementary Fig. 2). Figure 3a summarizes results using one of the DIP-α/Dpr10 crystal conformations (chains C and D of the asymmetric unit). The figure shows an alignment of the four purple Dprs with blue Dpr10. Each column in the alignment is color-coded based on the ΔΔG for mutating a residue in Dpr10 to the aligned residue in a purple Dpr.

We identified three interfacial positions, 10, 29, and 31 (Fig. 3a), as containing residues that function as negative constraints. Notably, stabilizing mutations (shown in green) suggest that DIP/Dpr complexes are not fully optimized for binding (see below). The destabilizing effect of the different negative constraints can be understood in biophysical terms (Fig. 3b). Dpr10 has a conserved hydrophobic residue at position 10 whose interaction with buried hydrophobic residues of DIP-α contributes favorably to binding (first panel in Fig. 3b). When this interaction is lost via mutation to glycine (Dpr11 and Dpr15), a negatively charged glutamate (Dpr16), or a polar asparagine (Dpr17), the DIP/Dpr complexes are destabilized either through the creation of a cavity in the hydrophobic interface or from the desolvation penalty associated with placing charged or polar atoms in the hydrophobic environment. Two other negative constraints correspond to conserved charged Glu and Lys at interfacial positions 29 and 31. These residues form salt bridges in the purple DIP/Dpr subfamily, but would create unsatisfied charges in the context of binding to DIP-α.

We tested these predictions with SPR measurements of wild type and mutant proteins. At least five independent measurements for each of three wild type interactions, Dpr10/DIP-α, Dpr6/DIP-α, and Dpr11/DIP-γ, resulted in average $K_D$s of 1.5 ± 0.1 μM for (15% experimental error), 2.0 ± 0.2 μM for (10% of experimental error) and 7.9 ± 0.9 μM for (11.4% experimental error), respectively. A similar level of experimental error (10–15%) is expected for the $K_D$s of all other Dpr or DIP mutants determined in this study, where each concentration was tested in duplicate but in a single experiment. Fitting errors and experimental errors are displayed in each figure containing SPR data.

Figure 3c displays SPR sensorgrams of Dpr10 wild-type and its mutants passed over a chip immobilized with DIP-α. As can be seen in the figure, the V31K single mutant increases the $K_D$ relative to wild-type by about an order of magnitude; adding a second mutation, Q29E, further increases the $K_D$ by about a factor of 2 while adding a third mutation L10G results in a triple mutant with a $K_D > 500$ μM. Thus, introducing Dpr11-like negative constraints to Dpr10 at these three positions is sufficient to kill binding between Dpr10 and DIP-α. We then carried out a reverse set of experiments aimed at removing negative constraints on

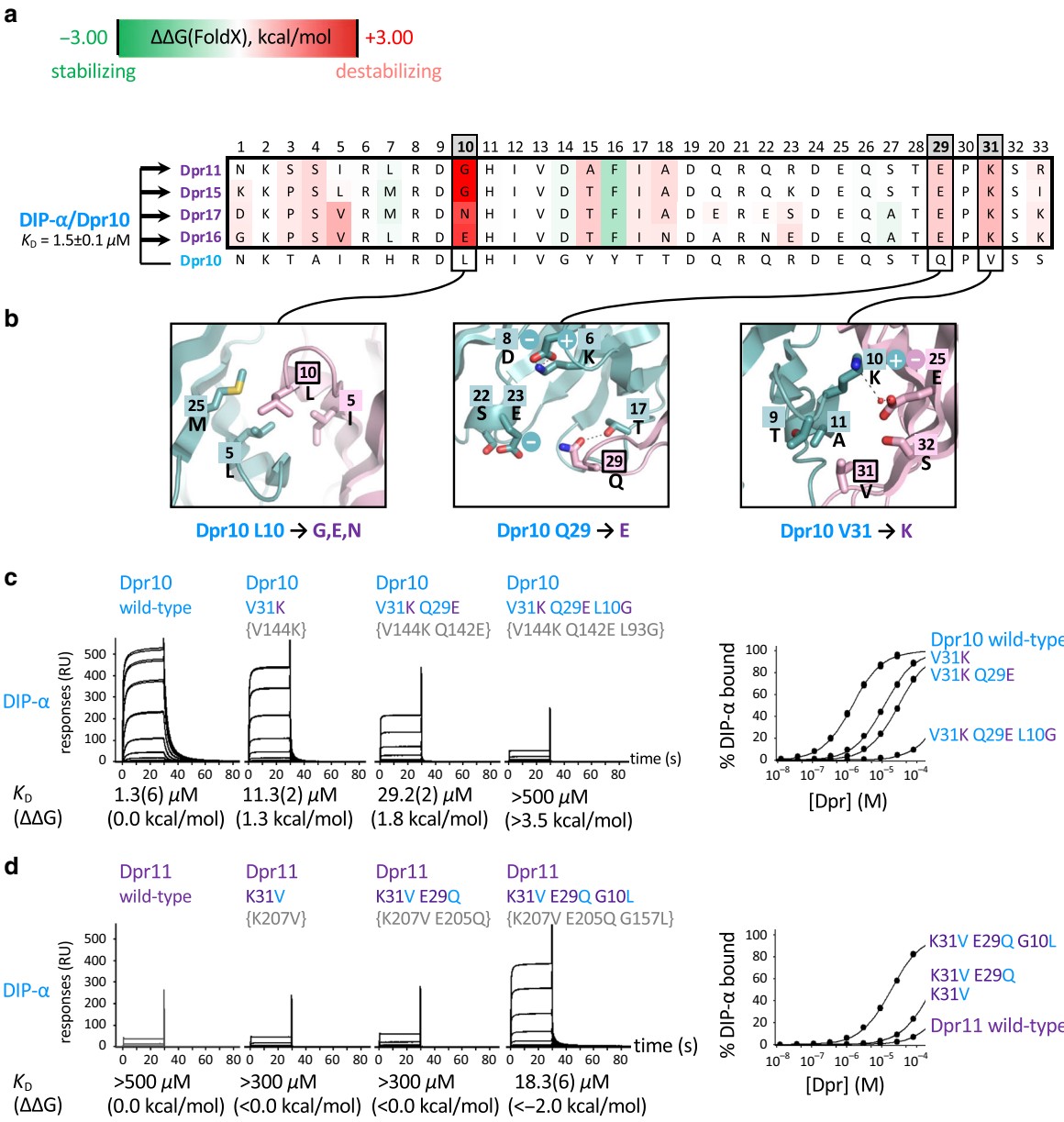

**Fig. 3 Negative constraints on purple Dprs 11/15/16/17 which prevent binding to the blue DIP-α. a** Alignment of interfacial residues of the four purple group Dprs and of Dpr10. Residues are color-coded based on the calculated ΔΔG, as specified in the color bar at the top of the figure. Predicted negative constraint positions are in gray. The average $K_D$ and the experimental error for wild-type DIP-α/Dpr10 interaction based on six independent experiments is given on the left of the alignment. All calculations are carried out on the DIP-α/Dpr10 complex indicated to the left of the figure. The arrows indicate that residues in Dpr10 were computationally mutated to those of the four purple Dprs, as indicated. **b** Structural details of wild-type environment for each position where a mutation is predicted to cause a negative constraint (enclosed in a bold box). **c, d** SPR binding analysis of Dpr10 and Dpr11 mutants, representing negative constraints, to DIP-α. Each row shows SPR sensorgrams of Dpr analytes binding over a DIP-α immobilized surface. An overlay of the binding isotherms for each surface is shown to the right. $K_D$ (ΔΔG) values for each DIP/Dpr interaction are listed below the SPR sensorgrams. For each $K_D$, the number in parenthesis represents the fitting error in the last significant figure, in μM for a single experiment with an expected experimental error up to 15%. Family-wide numbering of interfacial positions is used for all the mutants. Uniprot numbering is given in curly brackets. Source data are provided as a Source Data file.

Dpr11 to enhance its binding to DIP-α. As can be seen in Fig. 3d, the K31V single mutant and the K31V,E29Q double mutant have very little effect, while the triple Dpr11 mutant binds to DIP-α with a $K_D$ of 18.3 μM—weaker than wild-type Dpr10 but still quite strong. Overall, our results indicate that we have successfully identified the negative constraints on the Dpr side of the interface that preclude binding of purple group Dprs to blue group DIPs.

b. We identify positions 6,9,11,15,22 as containing negative constraints on the blue subgroup DIP-α that weaken binding to purple subgroup Dprs (Fig. 4a). SPR measurements showed that the residues at positions 9, 15, and 22, all weakened the binding of DIP-γ to Dpr11, although only position 9 had a substantial effect, increasing the $K_D$ of DIP-γ/Dpr11 binding from 7.9 to 37.2 μM (Supplementary Fig. 3A). As can be seen in Fig. 4a, there is a charged residue

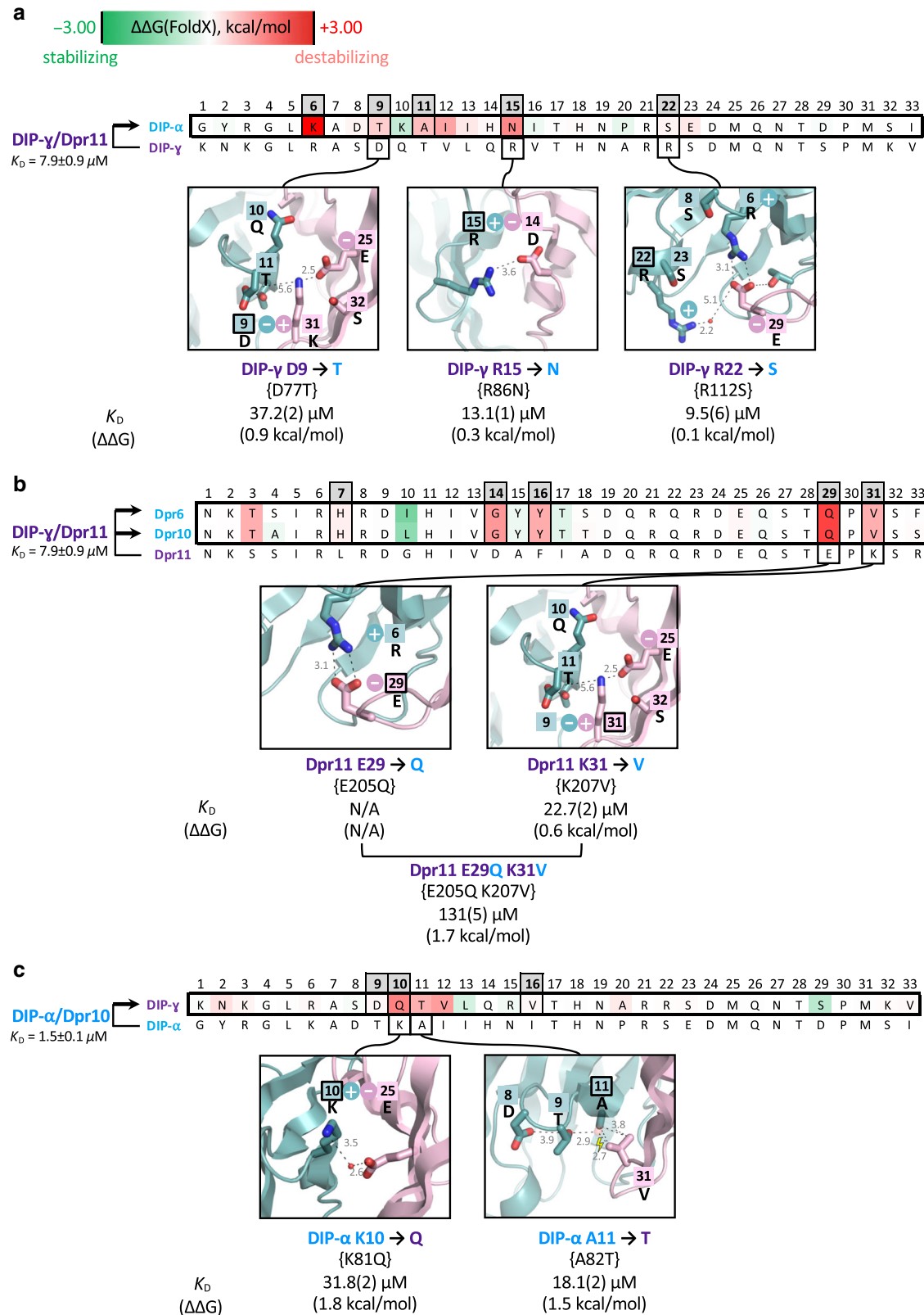

at each of these positions in DIP-γ that forms an ion pair with a complementary charge on purple group Dprs. Each of these positions contains a neutral polar residue in DIP-α which would then disrupt the ion pair and leave an unsatisfied partially buried charge in a hypothetical complex with purple group Dprs.

c. Positions 7,14,16,29,31 on Dpr6 and Dpr10 are predicted to contain residues that serve as negative constraints on blue subgroup Dprs that weaken binding to purple subgroup DIP-γ (Fig. 4b). Positions 29 and 31 were tested and, as can be seen in the figure, the single K31V mutant increased the $K_D$ of wild-type Dpr11 binding to DIP-γ from 7.9 to

**Fig. 4 Negative constraints on DIP-α, blue Dprs 6/10, and DIP-γ that inhibit purple/blue inter-subgroup binding. a** Negative constraints on DIP-α. Alignment of interfacial residues of DIP-α and DIP-γ followed by structural details used to explain negative constraint. The average $K_D$ and the experimental error for wild-type DIP-γ/Dpr11 interaction based on five independent experimental measurements is given on the left of the alignment. Experimental $K_D$s and ΔΔGs relative to wild type DIP-α/Dpr11 are shown below each insert. All symbols as in Fig. 3. **b** Negative constraints on Dprs 6/10. Alignment of interfacial residues of Dpr6 and Dpr10 with Dpr11. All other details as in **a**. **c** Negative constraints on DIP-γ. Alignment of interfacial residues of DIP-α and DIP-γ. All other details as in **a**. The supporting SPR data for the mutants presented in **a–c** can be found in Supplementary Fig. 3A–C. For each $K_D$, the number in parenthesis represents the fitting error in the last significant figure, in μM for a single experiment with an expected experimental error up to 15%. Source data are provided as a Source Data file.

22.7 μM while the K31V,E29Q double mutant with two negative constraints had a $K_D$ of 131 μM, almost a factor of 20 greater than wild-type (Supplementary Fig. 3B). The source of the destabilization is evident from the structure figures shown in Fig. 4b. The K to V mutation at position 31 would create a steric clash with Thr at position 11 of DIP-γ and leave two unsatisfied negative charges in the DIP-γ/Dpr11 interface (Asp9 of DIP-γ and Glu25 of Dprs), one of which will also be in a hypothetical complex with blue Dprs (Glu25). The E29Q mutation breaks a salt bridge with Arg6 of DIP-γ leaving an unsatisfied positive charge.

d.  Residues at positions 9, 10, and 16 in DIP-γ are predicted to serve as negative constraints on the purple subgroup DIP-γ that weaken binding to blue subgroup Dprs based on energy and evolutionary filters (Fig. 4c). Position 11 would also be predicted but only two of the three crystal structures used as templates yielded positive values for ΔΔG. The third corresponded to one of the two recently determined crystal conformations of DIP-α/Dpr10[10] reported after experimental validation of this position (Supplementary Fig. 3D). Mutations at Dpr10 positions 10 and 11, K10Q and A11T, were tested for binding to DIP-γ and each increased the $K_D$ of binding by about an order of magnitude, thus confirming the prediction. Again, the physical basis of the negative constraint is evident from the structure: K10 forms a water-mediated salt-bridge with sequence invariant Glu at position 25 of light blue Dprs that would be disrupted with a neutral Gln; The A to T mutation at position 11 of DIP-γ would introduce a steric clash with Val31 of light blue Dprs.

**Negative constraints between blue and green subgroups**. In a previous study, we tested mutants designed to switch the binding preference of the blue subgroup Dpr6 from its cognate DIP-α to the green subgroup DIP-η and, in turn, switch the binding preference of the green subgroup Dpr4 from DIP-η to DIP-α[11]. The relevant SPR data from that study are compared to our current FoldX predictions in Supplementary Table 3. As shown in the table, Dpr6 mutations at positions 7 and 31 weaken binding to DIP-α (Supplementary Table 3A) while reverse mutations at these positions in Dpr4 increase binding to DIP-α (Supplementary Table 3B). These results are in good qualitative agreement with the FoldX calculations and clearly indicate that positions 7 and 31 in blue subgroup Dprs serve as negative constraints that weaken binding to green subgroup DIPs. On the other hand, the SPR data show that position 31 in green Dpr4, but not position 7, contains a negative constraint that weakens binding to its cognate partner DIP-η (Supplementary Table 3C) and, consistently, a mutation at position 31, but not 7, increases binding to non-cognate blue DIP-α (Supplementary Table 3D). As can be seen in the table, these experimental results are also in good agreement with FoldX calculations.

**DIP/Dpr family overview**. Figure 5 displays negative constraints identified on the surface of hypothetical interfaces that would be

formed by a number of non-interacting subgroups. Most of the interface is used to create negative constraints but different regions are exploited by different subgroups (Fig. 5). Overall, our predictions implicate 21 positions on the Dpr side of an interface and 17 positions on the DIP side as a source of negative constraints (see positions highlighted in gray, Fig. 6). These positions are buried to different degrees in the interface, are located both in loops and secondary structure elements and have different physical–chemical properties (hydrophobic, polar, or charged). The majority of the negative constraints reside in loop regions and code for polar or charged residues (21 positions) with 10 positions burying more than 50 Å$^2$ of accessible surface area upon complex formation. Each non-cognate DIP/Dpr subfamily pair was found to be destabilized by a set of about 7 interfacial residues involving both sides of the interface. The physical source of the negative constraint can be shape non-complementarity in the hydrophobic core (Fig. 2) (through steric clashes or cavity creation), electrostatic effects (Coulomb repulsion or a desolvation penalty associated with burying unsatisfied charges or polar residues), or both (see examples in Fig. 5). Notably, our SPR results suggest that one negative constraint is not sufficient to ablate binding but, at least two are required to assure that members of different subgroups do not form a stable complex.

Our structure-based predictions of negative constraints include all positions predicted by sequence-based methods, but identify about twice as many positions. Sequence-based methods often fail to identity positions not fully conserved within a subfamily[22,35]. We also note that the majority of sequence-based methods encounter difficulties in identifying sites that are specific to only one subfamily. For example, positions 14 and 16 correspond to conserved Asp and Phe residues in the purple Dpr subfamily, while other Dprs have Gly and Tyr in these positions, respectively; see Fig. 2). Similarly, we predict several residues in the hydrophobic core as potential negative constraints (at positions 12, 13, 16 in Dprs and 5, 13, 16 in DIPs), whereas sequence-based methods predict only one of these positions (Fig. 2). Of note, these positions were not among those we experimentally confirmed in this study.

**Balancing affinity and specificity**. In some interfacial positions, FoldX predicts that mutations to residues found in other subgroups would actually increase affinity (Figs. 3, 4 and Supplementary Data 1). As an example, position 10 in purple Dprs, which is either a Gly or a polar residue, is predicted by FoldX to disrupt nonpolar interactions with DIPs in other subgroups. SPR experiments confirmed this prediction (Supplementary Fig. 4); L10G in Dpr10 decreased binding to DIP-α and DIP-β (ΔΔG > 1.6 kcal mol$^{-1}$), Notably, G10L increased cognate binding between Dpr11 and DIP-γ by about the same amount, indicating that some cognate binding affinity has been "sacrificed" so as to weaken non-cognate binding.

In a second example, position 7 is predicted to be used by the green Dpr subfamily to weaken non-cognate binding to DIP-α by placing an unsatisfied charge, K7, in non-cognate interfaces (Fig. 5). Confirming this prediction, the H7K {H110K} mutation in Dpr6

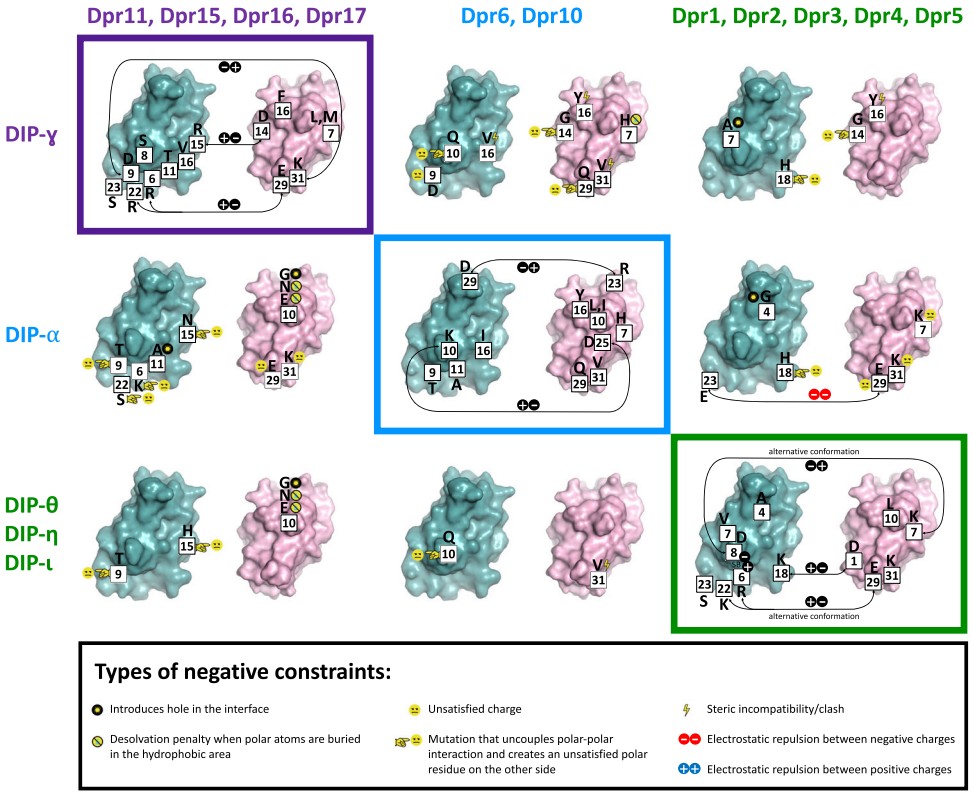

**Fig. 5 Negative constraints in DIP/Dpr non-cognate pairs.** Negative constraints in three DIP/Dpr subfamilies predicted using the protocol are shown on the DIP (in cyan) and Dpr (in pink) interacting surfaces of purple, blue, and green subfamilies using an open book representation, with cognate interacting pairs shown enclosed in boxes coded with subfamily color. Family-wide interfacial position number are in filled white boxes. Amino acid residue identity at a negative constraint position is given in a one letter code. The physical origin of each negative constraint is shown with a pictogram as detailed in the enclosed rectangle at bottom.

weakens cognate Dpr6/DIP-α binding by >1.9 kcal mol$^{-1}$ (ref.[11]). By contrast, the K7H {K82H} mutation in Dpr4 strengthens cognate binding to DIP-η by 0.3 kcal mol$^{-1}$ (ref.[11]), a small effect but outside the error bars of our measurements. This is another case where cognate binding appears designed to be suboptimal.

## Discussion

We have described the structural and energetic origins of the partition of DIPs and Dprs into orthogonal specificity groups defined by SPR-derived binding affinity measurements. We previously analyzed specificity determinants in type II cadherins[5], nectins[8], and DIPs and Dprs[11], primarily through visual inspection of sequences guided by structural data. Here we have adopted a far more extensive and quantitative approach as required by the complexity of the problem we set out to address. Specifically, how specificity is coded on seven subgroups of DIPs and Dprs so that out of the 49 possible subgroup pairings of closely related subfamilies of DIPs and Dprs, only seven form strong pairwise interactions. The problem is complicated by the fact that each subgroup contains between 1 and 5 members on the Dpr side and 1 and 3 members on the DIP side. Asking how 42 combinations of DIP and Dpr subfamilies are designed not to interact, or in some cases to interact weakly, poses a unique set of challenges. Notably, sequence-based methods were able to provide only partial answers to this question.

The approach we have adopted is essentially to build a large set of homology models of complexes that form and do not form and to ask, using FoldX calculations, SPR measurements and visual inspection what is wrong with those that do not form. Overall, we identified 38 positions which are used for negative constraints and carried out 23 validation tests with SPR at 8 of those positions (more than one mutation was tested at each position). In addition, our predictions are in agreement with previously published SPR experiments on the blue and green subfamilies[11] (Supplementary Table 3); three predicted negative constraints were confirmed and one prediction that a residue was not a negative constraint was also confirmed .

The negative constraints we identified for the most part exploit strong localized unfavorable energetic contributions rather, for example, than more subtle effects that are distributed over many residues (as is the case for type I cadherins where entropic effects involving the movement of entire domains play a role in determining dimerization affinities)[7]. The physical origins of negative constraints include: replacing a charge in an ion pair with one of opposite sign leading to Coulombic repulsion; replacing one member of an ion pair with a neutral group, which has the effect of burying an unsatisfied charge; mutations in the hydrophobic core to larger amino acids that create steric clashes or to smaller ones that create cavities which lead to packing defects and to weaker hydrophobic contributions to binding. Of course, the same "trick" cannot be reused for 42 different subgroup pairs. In fact, every set of negative constraints on the DIP and Dpr side of the 42 non-cognate pairs is unique in the sense that the same set is not reused (Supplementary Table 4). This is why as discussed above, most of the interface needs to be exploited. Notably, in cases we tested[11], at least three negative constraints had to be added or removed to essentially switch the specificity of a particular protein from one subgroup to another. As can be seen in Fig. 1b, intra-subgroup $K_{D}$s are generally in the 10 μM range

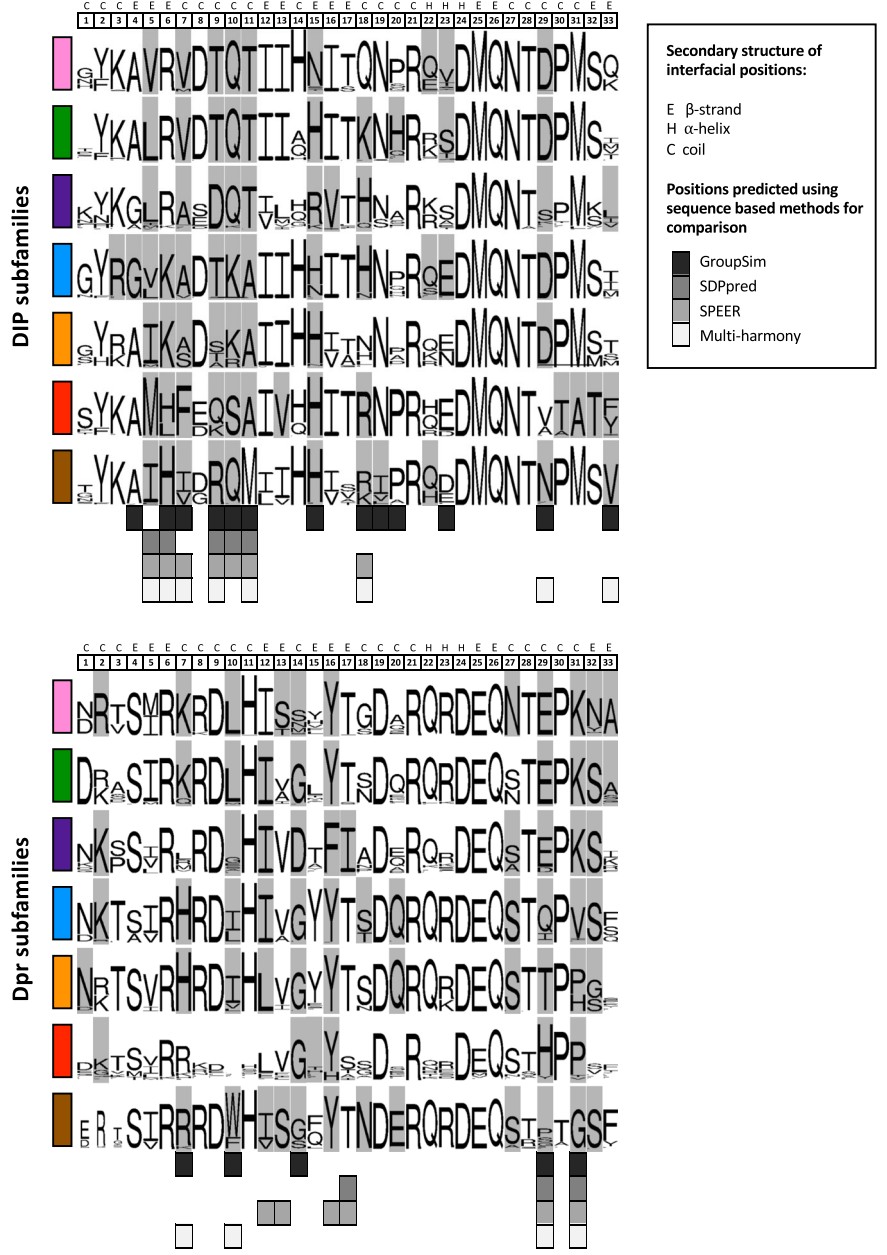

**Fig. 6 Predicted positions of negative constraints in DIP/Dpr subgroups.** Logos created for *Insecta* with residues that function as negative constraints highlighted in gray. Subfamily color-code as in Fig. 1b. Each interfacial position (1–33) is annotated above logos with associated secondary structure, as detailed in the boxed legend. Sequence-based predictions of specificity determining positions using GroupSim, SDPpred, SPEER, and Multi-Harmony methods (same as in Fig. 2a) are given below logos as specified in boxed inset.

(although some are as weak as 40 μM) while inter-subgroup $K_D$s (with a few exceptions) are undetectable, implying a $K_D > 500$ μM. This suggests that the generation of a new DIP/Dpr specificity requires a change in binding free energy of ~2.5 kcal mol$^{-1}$, a value that is difficult (but not impossible) to reach with one or two mutations.

How is the DIP/Dpr interface designed to achieve affinity and specificity? First, we note that, in common with most cell-cell recognition proteins, DIP/Dpr affinities are in the μM range, a thousand-fold weaker, for example, than many antibody-antigen complexes with substantially smaller binding interfaces. That lower affinities are a general feature of adhesion receptors, even those which are not members of large families, suggest that there are other factors involved that have little to do with negative constraints, for example cellular motility. Moreover, cell–cell

avidity involves the interactions of multiple receptors so that there may not be a need to evolve proteins with nanomolar affinities. This may account for the observation above that even cognate interfaces are not optimized for affinity. Moreover, a trade-off between affinity and specificity is to be expected in large protein families with only limited sequence divergence. If an interaction between two family members, say "a" and "b", is very strong, then it is more of a challenge, for example, to generate an isoform "c" that is similar in sequence to "b", but interacts very weakly with "a". Thus, the higher the specificity requirements, the greater the constraints on absolute affinities. That some residues in the interface are not optimized for cognate binding (see above) may then reflect the need to ensure that binding is not too tight.

We find that more than a half of the DIP/Dpr interface (38 of 66 interfacial residues) is utilized for negative constraints across

different insect species. However, these only play a role in the context of non-canonical subgroup–subgroup interactions, whereas in any given cognate DIP/Dpr complex, a majority of the interfacial residues likely play a stabilizing role. One can imagine that an ancestral DIP/Dpr complex relied entirely on stabilizing interactions in the hydrophobic core along with additional stabilizing or energy neutral interactions in the periphery. As multiple family members evolved the need for specificity was satisfied through the introduction of negative constraints throughout the interface. This evolutionary strategy is likely used by other large protein families. Of note, it has been found that specificity can be achieved through the insertion and/or deletion of loops[36] but this strategy does not appear to have been used by DIPs and Dprs.

Detailed binding affinity measurements have now been reported for a number of families of adhesion receptors. In some cases, for example type II cadherins[5], the family can be divided into subgroups whose members exhibit considerable promiscuity in their intra-subgroup binding properties while not binding to members of other subgroups. This behavior is reminiscent of the subgroup characteristics of DIPs and Dprs although, since there are fewer cadherin specificity groups, their evolutionary design is less of a challenge. In both cases binding promiscuity may well be a source of functional redundancy but this has not been established. By contrast, Dscams[1,2] and clustered protocadherins[3,4] are strictly homophilic which, as mentioned above, is an absolute requirement for their role of establishing a unique identity for every neuron. In contrast to DIPs and Dprs, the exquisite specificity of these protein families is established through multidomain interfaces that allow for a greater degree of finetuning[2,37–41]. This could result from simply a larger surface area on which to place negative constraints and/or from constraints[39] imposed by a multi-domain structure. Negative constraints are obviously a central component of evolutionary design in Pcdh and DSCAM protein families, but these have not yet been characterized in detail.

The range of DIP/Dpr binding affinities indicated in Fig. 1b raises an additional set of interesting questions regarding the biological role of weak interactions. In a number of cases there is a clearly established connection between molecular affinities and cellular phenotype[8,42] but in general very little is known about this topic. The situation becomes more complex for cells that express more than one family member and/or more than one subgroup member. The ability to design mutants of altered affinities and specificities combined with modern gene editing techniques opens the door to a new type of experiment which probes the molecular basis of cell-cell recognition. We have reported preliminary studies on this problem involving DIP and Dpr mutants which abolished homophilic or both homophilic and heterophilic interactions in the *Drosophila* retina[11,43]. The next logical step will be to test how subtle changes in affinity and specificity would affect biological phenotype.

## Methods

**Protein structure alignment.** The DIP/Dpr structures used in alignment presented in Fig. 1c were, blue subgroup—crystal structures of Dpr6/DIP-α (PDBID: 5EO9) and two conformations of Dpr10/DIP-α (PDBID: 6NRQ)[9,10]; purple subgroup—crystal structure of Dpr11/DIP-γ (PDBID: 6NRR)[10]; green subgroup—crystal structures of Dpr4/DIP-η (PDBID: 6EG0) and for two conformations each for Dpr1/DIP-η (PDBID: 6NRW) and Dpr2/DIP-θ (PDBID: 6EG1)[10,11]. The structures were aligned and visualized using PyMOL Molecular Graphics System, Version 2.0 Schrödinger, LLC.

**Construction of homology models for all superfamily members.** We built homology models for each DIP and Dpr Ig1 domain (using the crystal structure of the DIP-η/Dpr4 complex as a template, PDBID: 6EG0). The models are expected in principle to be high quality since there is considerable sequence and structural similarity between intra-family DIP and Dpr Ig1 domains. A few modeled proteins have longer loops than those of the template but these regions do not appear

critical for Ig1–Ig1 interface. Homology models of DIP and Dpr monomers built using MODELLER[44] were superimposed onto Dpr4/DIP-η crystal structures yielding 231 DIP/Dpr heterodimers. The side chains of the dimer models were further minimized using the Scwrl4 algorithm[45].

**FoldX protocol to evaluate ΔΔG (binding).** Although FoldX can be used as a "black box" we summarize here the procedures used to calculate ΔΔG(binding). We first refine the structure of a protein complex using the FoldX utility called "RepairPDB" (http://foldxsuite.crg.eu/command/RepairPDB). This procedure "repairs" improbable dihedral angles and van der Waals clashes in a protein. We then run five rounds of "RepairPDB" and confirm that total energy of a protein complex reaches a plateau. The complex whose structure has now been optimized with respect to the FoldX energy function is then used as a starting point for a computational mutagenesis procedure with "BuildModel" (http://foldxsuite.crg.eu/command/BuildModel). A desired mutation is made to the repaired structure and then energy-minimized by sampling rotamers for the mutated residue and its neighboring amino acids. The wild type protein is then energy-minimized with respect to rotamers of the same set of residues. The difference in interaction energy between the mutant and wild type is then calculated between the two protomers in the complex using the "AnalyseComplex" FoldX utility (http://foldxsuite.crg.eu/command/AnalyseComplex).

To test if FoldX achieves convergence for a given mutation we run "BuildModel" ten times by setting the "numberOfRuns" parameter to 10 (http://foldxsuite.crg.eu/parameter/numberOfRuns), which produces ten pairs of wild type and mutant structures, as different neighbors of the mutated residue could be moved during minimization. The average value of the interaction energy obtained via energy decomposition of the total energy using "AnalyseComplex" defines ΔΔG (binding).

Since we carried out ten separate runs for each FoldX prediction, this yields 30,440 FoldX single point mutation calculations (1 week of computing time on 1 CPU).

It is important to note that our homology models give comparable FoldX results to those obtained from crystal structures. This is perhaps not surprising since we use high-quality homology models based on high sequence identities and the fact that all DIP/Dpr crystal structures superimpose to within 1 Å. To illustrate this point, we carried out FoldX calculations on homology models and compared the results to those obtained from crystal structures of Dpr11/DIP-γ, Dpr6/DIP-α, and Dpr10/DIP-α. The Pearson correlation coefficient (and root mean square error) with experimental values was 0.56 (1.65) for the crystal structures and 0.53 (1.10) for the homology models based on 25 experimental measurements of binding energy differences.

**Calculating the effects of mutations on DIP/Dpr binding.** The following is a summary of the methods evaluated for the calculation of the effects of mutations on binding free energies. FoldX evaluates the effects of mutations using an empirical force-field that allows side chains to move but keeps the backbone rigid[18,31]; mCSM is a machine learning method based on the assumption that the impact of a mutation is correlated with atomic-distance patterns in surrounding amino acids[30]; BeAtMusic relies on a set of coarse-grained statistical potentials derived from known protein structures[27]. BeAtMusic is the fastest of all the methods we tested—it takes seconds to assess all possible mutations in a protein chain or at the interface; MutaBind utilizes a combination of molecular mechanics force fields, statistical potentials and a fast side-chain optimization algorithm; Rosetta flex ddG[24] samples conformational diversity using "backrub" to generate an ensemble of models and then applies torsional minimization, side-chain repacking, and averaging across this ensemble to calculate ΔΔG; BindProfX[32] combines a sequence profile conservation score of structural homologs with the FoldX potential. Data on the performance of these methods is summarized in Supplementary Table 2.

**Sequence-logo preparation.** To get orthologs of DIPs and Dprs in Insecta species we performed the search using reciprocal best hit BLAST[46] with an *e*-value cutoff of $10^{-35}$ using in-house developed scripts on non-redundant BLAST database. The resulting 2950 Dpr sequences and 2005 DIP sequences (121 species) were screened using CD-HIT[47] with 100% threshold to get rid of redundancy arising from multiple GI (GenInfo Identifier) numbers pointing to the same protein product. The obtained non-redundant set of full-length sequences corresponding to each DIP or Dpr was combined with a reference sequence (DIP-α or Dpr6 respectively), and aligned using Clustal-Omega[48]. The multiple sequence alignments (MSA) of each DIP or Dpr were further processed in Jalview[49] to remove sequences that had non-complete Ig1 domains and to extract interfacial positions based on the reference sequences. Sequence logos based on the MSAs presented in Fig. 2 were created using WebLogo[50] separately for each *Drosophila* DIP and Dpr based on 2570 Dpr and 1732 DIP sequences. Sequence logos were also prepared on a subfamily level by combining sequences of all its members (Fig. 6).

**Sequence-based methods used to predict specificity residues.** GroupSim[20] analyzes evolutionary conservation patterns using physico-chemical properties of amino acids by means of similarity matrices; SDPpred[21] uses information theory

schemes based on Shannon entropy and frequency scores; SPEER[22] uses amino acid properties, entropy and evolutionary rates by analyzing quantitative measures of the conservation patterns of protein sites based on their physico-chemical properties and the heterogeneity of evolutionary changes between and within the protein subfamilies. Multi-Harmony[23] uses a combination of Shannon entropy and machine learning.

All the sequence-based methods we use for determination of specificity positions require predefined subgrouping of sequences based on their binding preferences. We grouped the above 2570 Dpr and 1732 DIP sequences (truncated to 33 interfacial residues only) into seven sets (DIP-θ/η/ι/, DIP-κ, DIP-α, DIP-β/λ, DIP-γ, DIP-δ, DIP-ε,ζ for DIPs, and Dpr1/2/3/4/5, Dpr7, Dpr6/10, Dpr8/9/21, Dpr11/15/16/17, Dpr12, Dpr13,14,18,19,20 for Dprs).

We rely on the assumption that orthologs in the same subfamily conserve their interaction specificity. This assumption would not work for comparison of proteins in species that are too far removed from one another on the tree of life, but since we consider orthologs within one class, *Insecta*, the assumption is likely to be correct. To justify this, we calculated the average pairwise sequence identities within *Insecta* orthologs for interfacial residues of each DIP and Dpr. The average pairwise sequence identity for interfacial residues of DIP (86%) and Dpr (83%) orthologs in *Insecta* is 84%. This value is similar to the average intra-subgroup pairwise sequence identity for each DIP and Dpr *Insecta* subgroup (78%) and the intra-subgroup pairwise sequence identities of interfacial residues in *Drosophila melanogaster* (~80%). These values are also consistently higher than inter-subgroup pairwise sequence identities in *Drosophila melanogaster* or *Insecta* (~50%). Only one subfamily (red Dpr) is an exception to the rule—it has relatively low conservation both in *Insecta* (48%), and in *Drosophila melanogaster* (40%).

**Protocol for identifying negative constraints.** We applied "energy filter" by carrying out FoldX calculations on the effects of mutations on X-ray structures of DIP/Dpr complexes of *Drosophila melanogaster* where available and on homology models where crystal structures were not available. If mutations were predicted to cause destabilizing effects ($\Delta\Delta G \geq 0.05$ kcal mol$^{-1}$) for every DIP or Dpr fly subgroup member in the context of binding to a member of another Dpr or DIP subgroup, the position were accepted provided $\Delta\Delta G \geq 0.05$ kcal mol$^{-1}$ for every subfamily template complex used in the FoldX calculations. The cutoff value for $\Delta\Delta G$ was chosen based on the estimated standard deviation we found for mutating an interfacial amino acid into self ($\Delta\Delta G = 0.00 \pm 0.05$ kcal mol$^{-1}$). Positions that passed the energy filter were further subject to the evolutionary filter (see below).

We applied "evolutionary filter" by filtering positions using multiple sequence alignments (MSAs) for DIP and Dpr subfamilies based on sequences of insect orthologs (see above). For every non-cognate DIP-i/Dpr-j family, to filter positions on the DIP side we compared corresponding positions from DIP-i and DIP-j subfamilies in MSAs. To filter positions on the Dpr side we compared positions from Dpr-i and Dpr-j subfamilies in MSAs. The position was accepted if at least one of the three conditions outlined below was true.

1. If positions in "i" and "j" subgroup were both conserved (Shannon entropy, $S < 0.23$) and different in identity—the position was accepted. $S$ is given by

$$S = -\sum_{i=1}^{N} f_i \log_{21} f_i,$$

where $f_i$ is the fraction (frequency of occurrence) of residues of amino acid type present at a position in MSA and $N$ is the number of amino acid types (base of 21 was picked to account for 20 amino acids types and a gap).

2. If residues in the "i" and "j" subgroups had difference in their biophysical properties. We binned the 20 amino acids into four property groups: hydrophobic (I, V, L, M, F, W, A, C, G, P), polar (S,T,Q,N,H,Y), positively charged (K,R), and negatively charged (E,D). Then, for every amino acid at a position of subfamily "i" ($aa_i$) and for every amino acid at the same position of subfamily "j" ($aa_j$), we checked if $aa_i$ and $aa_j$ belonged to the same or different biophysical property groups defined above. When $aa_i$ and $aa_j$ were different in property, we included the contribution of the amino acid pair weighted by amino acid occurrences in MSAs ($f_{aa_i} \times f_{aa_j} \times 100\%$) to the overall percent difference. When $aa_i$ and $aa_j$ had the same property, the contribution was not included. The sum of all included weighted contributions ($\sum_{k,l}^{N} f_{aa_{ik}} \times f_{aa_{jl}} \times 100\%$, where $N$ is the number of amino acid pairs where (property of $aa_i$) ≠ (property of $aa_j$), while $k$ and $l$ run over all amino acids present at a position in subfamilies "i" and "j", respectively) defined the total biophysical property difference in %. The positions were accepted at >90% cutoff.

3. If residues in the "i" and "j" subgroups had differences in size (amino acids of one subgroup always larger or always smaller in size than those appearing in the other subgroup. We binned the 20 amino acids into five size groups, which were based on the volume of amino acids: tiny (G, A, S), small (C, D, P, N, T), medium (E, V, Q, H), large (M, I, L, K, R), and bulky (F, Y, W). We computed two percent differences—the percent difference when $aa_i$ belonged to a larger size group than $aa_j$ and the percent difference when $aa_i$ belonged to a smaller size group than $aa_j$ (relative size differences between groups were encoded in the group names as tiny < small < medium

< large < bulky). Each percent difference was computed as a sum of the weighted amino acid contributions ($\sum_{k,l}^{N} f_{aa_i} \times f_{aa_j} \times 100\%$, where $N$ is the number of amino acid pairs where (size of $aa_i$ group) ≠ (size of $aa_j$ group), while $k$ and $l$ run over all amino acids present at a position in subfamilies "i" and "j", respectively) when the condition above was satisfied (size($aa_i$) > size ($aa_j$) in the first case and size($aa_i$) < size($aa_j$) in the second case). The maximum of the two percent differences defined the total size difference in %. The positions were accepted at >90% cutoff.

**Plasmid construction and protein expression.** All proteins were produced by expression in human embryonic kidney cells (Thermo Fisher Scientific). DNA sequences encoding DIP and Dpr extracellular regions were amplified by PCR and sub-cloned into the mammalian expression vector VRC-8400 (ref.[51]) between the NotI and BamHI sites. All primer sequences used in this study are given in Supplementary Table 5. Top 10 *E. coli* cells (Thermo Fisher Scientific) were used for plasmid manipulations. Sequence boundaries for each DIP and Dpr were the same as those defined in Cosmanescu et al.[11] All sequences were preceded by the signal sequence of human binding immunoglobulin protein BiP (MKLSLVAAMLLLL-SAARA), and a kozak sequence (GCCACC). Constructs were followed by a C-terminal hexa-histidine tag. Point mutations were introduced using the Quick-Change method (Agilent). HEK293F cells were transiently transfected with each expression construct using the polyethylenimine method[52], and cells were expanded and grown in shake flasks in a $CO_2$ incubator for 3–6 days.

**Protein purification.** Conditioned media was equilibrated to 10 mM Tris-HCl pH 8.0, 500 mM NaCl, 3 mM $CaCl_2$, and 5 mM Imidazole pH 8.0 and incubated with $Ni^{2+}$ charged IMAC Sepharose 6 Fast Flow resin (GE Healthcare) for 1 hr at 25 °C. Resin was washed with at least 20 column volumes of buffer containing 10 mM Imidazole pH 8.0. SDS gel electrophoresis was used to detect which elution fractions contained the desired protein.

Proteins were further purified by size-exclusion chromatography (Superdex 200 HiLoad 26/60 or Superdex S200 Increase 10/300 GL; GE Healthcare) on an AKTA pure fast protein liquid chromatography system (GE Healthcare). Most proteins were stored in a buffer of 10 mM Bis–Tris pH 6.6 and 150 mM NaCl. DIP-α and its mutants were stored in a modified buffer (10 mM Bis–Tris pH 6.0 and 150 mM NaCl) due to stability issues. Most proteins were stored in a buffer of 10 mM Bis–Tris pH 6.6 and 150 mM NaCl. UV absorbance at 280 nm was used to determine protein concentration and verification of purity was determined by gel electrophoresis. Accurate molecular weights were determined through MALDI-TOF mass spectrometry at the Proteomics Shared Resource facility at Columbia University.

**Sedimentation equilibrium by analytical ultracentrifugation.** Experiments were performed in a Beckman XL-A/I analytical ultracentrifuge (Beckman-Coulter, Palo Alto CA, USA), utilizing six-cell centerpieces with straight walls, 12 mm path length and sapphire windows. Protein samples were dialyzed to 10 mM Bis–Tris pH 6.6, 150 mM NaCl. The samples were diluted to an absorbance of 0.65, 0.43, and 0.23 at a 10 mm path length and 280 nm wavelength in channels A, B, and C, respectively. Dilution buffer were used as blank. The samples were run at four speeds. Most proteins were run at 16,350, 26,230, 38,440, and 52,980×g. For all runs the lowest speed was held for 20 h and then four scans were taken with a 1 h interval, the second lowest held for 10 h then four scans with a 1 h interval, and the third lowest and highest speed measured as the second lowest speed. Measurements were done at 25 °C, and detection was by UV at 280 nm. Solvent density and protein v-bar were determined using the program SednTerp (Alliance Protein Laboratories). To calculate the $K_D$ and apparent molecular weight, data were fit to a global fit model, using HeteroAnalysis software package, obtained from University of Connecticut[53] (http://www.biotech.uconn.edu/auf).

**Surface plasmon resonance (SPR) binding experiments.** SPR binding assays were performed using a Biacore T100 biosensor equipped with a Series S CM4 sensor chip. To minimize artificial binding resulting from enhanced-avidity effects of oligomers interacting with an immobilized ligand, DIPs and their respective mutants, were consistently used as ligands rather than analytes, and immobilized over independent flow cells using amine-coupling chemistry in HBS pH 7.4 (10 mM HEPES, 150 mM NaCl) buffer at 25 °C using a flow rate of 20 μL min$^{-1}$. Dextran surfaces were activated for 7 min using equal volumes of 0.1 M NHS(N-Hydroxysuccinimide) and 0.4 M EDC(1-Ethyl-3-(3-dimethylaminopropyl)carbodiimide). Each protein of interest was immobilized at ~30 μg mL$^{-1}$ in 10 mM sodium acetate, pH 5.5 until the desired immobilization level was achieved. The immobilized surface was blocked using a 4-min injection of 1.0 M ethanolamine, pH 8.5. Typical immobilization levels ranged between 760-980 RU. To minimize nonspecific binding the reference flow cell was blocked by immobilizing BSA in 10 mM sodium acetate, pH 4.25 for 3 min using a similar amine-coupling protocol as described above. For each experiment where DIP mutants were immobilized to the chip surface, the wild type molecule was immobilized on an adjacent flow cell as a positive control.

Binding analysis was performed at 25 °C in a running buffer of 10 mM Tris-HCl, pH 7.2, 150 mM NaCl, 1 mM EDTA, 1 mg mL$^{-1}$ BSA and 0.01% (v/v) Tween-20. Analytes were prepared in running buffer and tested at nine concentrations using a three-fold dilution series ranging from 81 to 0.012 μM for data shown in Fig. 3c, d, and Supplementary Figs. 3A–C, 4, with the of exception Dpr10 binding to DIP-α A11T in Supplementary Fig. 3 C, which was tested at nine concentration ranging from 27 to 0.004 μM, prepared in a three-fold dilution series. Dpr6 was tested at eight concentrations ranging from 9 to 0.004 μM, prepared in a running buffer using a three- fold dilution series for data presented in Supplementary Fig. 1A, with the exception of experiments over the DIP-α I16A and DIP-α K10Q G4S surfaces where Dpr6 was used at ten concentrations ranging from 81-0.004 μM, using a three-fold dilution series. Dpr10 binding experiments in Supplementary Fig. 1B, was tested at nine concentrations ranging from 27 to 0.004 μM, prepared in a running buffer using a three- fold dilution series, with the exception of experiments over surfaces immobilized with DIP-α I16A, K10Q G4S and K10Q D29S where the analyte was used at ten concentrations ranging from 81 to 0.004 μM, using a three-fold dilution series. In each experiment, every concentration was tested in duplicate. Within each experiment, there two technical replicates starting with a single concentration series, where samples are tested in order of increasing concentration, followed by a repeat of the same concentration series, performed again from low to high concentration. During a binding cycle, the association phase between each analyte and the immobilized molecule was monitored for either 30 or 40 s as indicated by the plotted sensorgrams, followed by 120-s dissociation phase, each at 50 μL min$^{-1}$. At the end of the dissociation phase the signal returned to baseline thus eliminating the need for a regeneration step. The last step was buffer wash injection at 100 μL min$^{-1}$ for 60 s. The analyte was replaced by buffer every two or three binding cycles to double-reference the binding signals by removing systematic noise and instrument drift. The duplicate binding responses were fit globally, using an 1:1 interaction model and a single $K_D$ was calculated as the analyte concentration that would yield 0.5 $R_{max}$ (ref.[54]) and a fitting error. $K_D$s < 100 μM were calculated using an independent $R_{max}$. For $K_D$s > 100 μM, the $R_{max}$ was fixed to a global value determined by the $R_{max}$ of a different Dpr analyte tested over the same DIP surface during the same experiment that showed binding above 50% and therefore produced a more accurate $R_{max}$ (ref.[54]). For $K_D$ > 300 μM, a lower limit is listed since at the analyte concentrations used we could not measure accurate $K_D$s even when the $R_{max}$ is fixed. The data was processed using Scrubber 2.0 (BioLogic Software).

For several wild type interactions discussed in this manuscript, Dpr10/DIP-α, Dpr6/DIP-α, and Dpr11/DIP-γ, we have also determined $K_D$s from independent experiments (see the source data file). For the Dpr10/DIP-α binding pair, the $K_D$ from six independent experiments is 1.5 ± 0.1 μM, and similarly for the Dpr6/DIP-α, the $K_D$ from six independent experiments is 2.0 ± 0.2 μM. The $K_D$ from five independent experiments for Dpr11/DIP-γ is 7.9 ± 0.9 μM. The standard deviation in ΔΔG values computed based on $K_D$ measurements of the above three cases do not exceed 0.1 kcal mol$^{-1}$. Therefore, mutations resulting in ΔΔG (SPR) > | 0.2 kcal mol$^{-1}$ | are likely to be significant.

In a note added in proof, Cheng et al.[10] recently criticized $K_D$s we previously determined via SPR experiments[11], specifically for the interacting pairs Dpr6/DIP-α, Dpr11/DIP-γ, and Dpr1/DIP-η, for being consistently weaker than $K_D$s they reported. The following discussion explains how differences in experimental methodology likely explains these differences and raises issues about the approach used by Cheng et al.[10]. The underlying challenge in both sets of measurements is that some DIPs homodimerize, a problem that is discussed by Rich and Myszka in their 2006 review of SPR data[55]. We addressed this problem by immobilizing DIP proteins to the chip surface, using Dprs as analytes, to minimize artificially lower $K_D$s resulting from homodimers used as analytes. It has been suggested that an alternate "fix" is to use the free monomer analyte concentration in $K_D$ calculations[10]. However, the monomer/dimer equilibrium in the analyte can shift the moment the analyte is injected over a surface immobilized with a heterophilic binding partner, which changes the monomeric analyte concentration available for a heterophilic binding reaction over the course of the experiment. Therefore, this correction does not adequately solve the problem.

Moreover, to calculate the free monomer analyte concentration of DIP-η, Cheng et al. used size exclusion chromatography and SPR to calculate a homophilic $K_D$ of 23 μM and 14 μM, respectively[10], compared to our analytical ultracentrifugation-determined $K_D$ of 56.2 μM (ref.[11]). Given the reliability of AUC measurements, this highlights the problems associated with Chang et al.'s approach. Most importantly these numbers suggest that the differences in $K_D$ reported in both studies are due in part to the use of different biophysical techniques used to study these molecules.

In addition, for many of the results that differ between our studies and Cheng et al.[10], these authors relied on ECIA experiments with artificially multimerized molecules. As previously discussed in detail in Cosmanescu et al.[11] although ECIA can be used to detect interactions in a high-throughput fashion and without the need for purified proteins, it involves the use of multimerized forms of both the prey and the bait molecules, and thus introduces artificial avidity to amplify binding signals, enhancing the likelihood of positive detection, but masking the real binding affinities between interactants. In addition, the use of unpurified protein supernatants[10] could introduce experimental bias toward identifying interactions between molecules that are more easily expressed at higher levels compared to proteins that can be difficult to express.

**Reporting summary**. Further information on research design is available in the Nature Research Reporting Summary linked to this article.

## Data availability
The raw data underlying Figs. 3, 4 and Supplementary Figs. 1–4 are provided as a Source Data file. PDB accession codes for structures used in this study: 5EO9, 6NRQ, 6NRR, 6EG0, 6NRW, and 6EG1. All other relevant data is available from the corresponding authors upon reasonable request.

## Code availability
The code used to filter negative constraints is available from the corresponding authors upon reasonable request.

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

## Acknowledgements

This work was supported by NIH grants R01 GM30518 (B.H.), R01 MH114817 (L.S.) and NSF grant MCB-1914542 (B.H.). We thank Engin Ozkan for providing DIP-α/Dpr10 and DIP-γ/Dpr11 structures and density maps prior to their appearance in the PDB database.

## Author contributions

A.P.S., L.S., and B.H. conceived the project, designed experiments, and analyzed data. A.P.S. carried out all the theoretical and computational work. P.S.K. performed and analyzed SPR experiments. G.A. performed AUC experiments. J.J.B., F.C., and S.M. produced expression vectors and proteins. A.P.S. and F.C. designed mutant constructs. A.P.S., B.H., and L.S. wrote the manuscript.

## Competing interests

The authors declare no competing interests.
