## [Peer Review File · Nature Communications]

Reviewers' comments:

Reviewer #1 (Remarks to the Author):

Sergeeva et al. report a detailed computational analysis of determinants of interaction specificity in *Drosophila* DIP and Dpr cell-cell recognition proteins. This system is intriguing because 7 subfamilies of proteins form preferred heterotypic interactions (DIP-to-Dpr protein interactions) without extensive cross-talk between subfamilies. The question addressed by the authors is how this intricate interaction specificity network is established at the structural level, which has been addressed before, but less comprehensively. Here the authors approach this question by building and analyzing homology models of numerous complexes and mutant complexes and predicting residues that can change binding specificity. They test selected predictions using binding experiments. An in-depth study of one pair of subfamilies includes experiments that confirm predictions of mutations that disrupt cognate complexes, as well as a nice example that shows how swapping appropriate residues into a weakly interacting complex can restore affinity. A fairly compelling picture emerges of how sets of distributed interactions are used to disfavor non-cognate DIP/Dpr complexes. The summary of the negative design strategies in Figure 5 provides a good overview. Overall the work is interesting, well done and makes a valuable contribution – by suggesting and illustrating general principles – that will shape thinking in the protein interaction field. There are some aspects of the paper that require clarification and the manuscript should be accompanied by a comprehensive report of the results of the analysis.

Three points related to Figure 1b. (1) Are these all fly proteins? (2) I couldn't find an explanation of where the binding data in that figure panel come from. Are these values all from reference 11? (3) The authors appear to assume that orthologs in the same subfamily conserve their interaction specificity, but they don't discuss why this is valid. This is a key point for the sequence-based analysis.

As written, it is confusing whether the members of a subfamily, when used for a given purpose, include only fly proteins vs. all orthologous insect proteins. E.g. when residues were mutated "to all residues appearing at the same position in DIPs of other specificity subgroups," did this consider only fly proteins? Lines 183 – 193 were particularly confusing in this regard. I am guessing there were two filters required to designate a residue as a negative constraint: (1) if analyzing fly subfamily *i*, the mutation must be predicted as disruptive when modeled in each fly protein member of subfamily *i* (giving $DDG > 0$ for at least one other, non-*i*, subfamily). (2) In a separate test, "we only defined a constraint as a position that would also be predicted for other Insecta species so as to focus on constraints that are evolutionarily conserved." I gather this second prediction was not based on structural modeling, but rather on a qualitative comparison of biophysical properties, and applied to a much larger number of proteins (which makes sense; the residue equivalence categories used should be reported in the methods). Regardless of whether this is the correct interpretation of what was done, the authors should make this section easier to follow (e.g. by liberally adding the use of "fly" as a qualifier, in many places in the paper, and making clear there were 2 different tests at this stage).

Then, on line 240 another criterion for a negative constraint is introduced: "In addition to this requirement, predictions are then filtered based on the requirement that $DDG > 0$ for all available blue subgroup crystal structures." Along with the statement on line 220 that "Calculations are carried out on every complex for which a crystal structure is available." A general question: Were two different protocols used, one to study all fly DIP/Dpr subfamily pairs (using just one or two templates each) and a different one for complexes involving the blue/purple groups (using all available crystal structures and adding at least one new filter)? Or was there just one protocol that is described slightly differently in different places? If there were two different protocols, did the results differ? This was confusing.

The paper should be accompanied by a list of the predicted negative constraints for all subfamilies, according to the described algorithm. Figure 6 highlights the positions that were found, but it would be good to include a comprehensive annotation (in the supplement) of which residues are predicted to destabilize which subfamily interactions. Or, even better, to make figure 5 for all subfamilies (given that the data are available). The paper says "Overall, we identified negative

constraints for 42 combinations of non-interacting DIP and Dpr subgroups (see supplemental excel file for all FoldX data)," but I don't see the list of negative constraints that passed the filters, nor was there an excel file among the materials I saw.

Line 133 states "We tested the ability of a number of sequence-based methods to identify specificity determinants in the seven specificity subgroups." And line 153 says "Below we compare the predictions of these sequence-based methods to an energy-based analysis of DIP and Dpr structures." To understand the result of that test and comparison, it should be easier for the reader to see which of these sequence-based predictions overlap with the structure-based predictions and which were experimentally verified in this work (or in other work). This could be added to Figure 6.

The paper states: "Similarly, we predict several residues in the hydrophobic core as potential negative constraints (12, 13, 16 in Dprs and 5, 13, 16 in DIPs), whereas sequence-based methods predict only one of these 344 positions (Figure 2)." But these were not among the mutations that were experimentally confirmed to disrupt binding (I don't think), which is worth mentioning here, since this may not be borne out.

Experimental error estimates are lacking. I did not find descriptions of experimental error or fitting error, other than for 2 replicates of the AUC measurements. Were the SPR experiments performed more than once? (The checklist indicates replicates, but the paper does not, as far as I saw). For Kd values, in Figure 3C and elsewhere, is the value in parentheses the error? E.g. 17.61(7) kcal/mol. What kind of error? In what units? How many replicates? Is this meant to indicate std. dev. of 0.07 kcal/mol (which seems way too low of an error for true replicates)? Related to this, many of the binding curves do not saturate. How was the 100% bound signal estimated, and is this well justified? The authors could report more conservative estimates of their measurement accuracies that would still substantiate their claims.

Minor points:

It would be helpful to clarify why there is a stochastic element of the FoldX modeling procedure. E.g. to explain why 10 runs gave different values.

Several different and contradictory values for the dissociation constant of DIP-g/Dpr11 are reported (7.1, 7.9, 8.5) in the text and figures.

Evaluating mutational energies by comparing mutant and WT structures with the same rotamers in each structure is an unusual procedure (though I can imagine it may cancel errors). Can the authors explain/justify this? And explain how the structure that was used was selected? I.e., from the methods: "This requires initial sampling of different rotamers for a mutated residue and subsequent synchronous movement of neighboring residues in a WT complex and a MT complex to find the optimal position for the mutated residue." But what is "optimal" in this context, given there are two different structures to consider?

I didn't understand this expression: $33 \times [3(10+19) + 1(10+17) + 177 \cdot 5(8+16) + 1(10+20) + 1(9+18) + 1(10+20) + 1(9+16)] = 11418$ mutations. First, is the "3" in "3(10+19)" meant to be "2" (for "two conformations of Dpr10/DIP")? Next, if the numbers like 10, 17, 8, 16, etc. are the numbers of proteins in non-i subgroups, this seems like an overestimate of the number of mutations to be made, given that many proteins share the same interface residue at a given site. It would be good to clarify the source of the terms in this expression.

Reviewer #2 (Remarks to the Author):

This is a very nice paper that explores the basis of binding specificity using an example of a physical interaction between proteins from DIP and Dpr families. The authors performed *in silico* mutational screening and found that some amino acid substitutions serve as negative constraints and prevent certain undesired interactions between members of these families. Moreover, the authors went further and performed experimental binding affinity measurements to confirm their computational findings. I have several suggestions which can potentially improve the paper:

1. It might be worth discussing that stabilizing mutations can also represent negative constraints since the studied interaction is rather weak and transient.

2. In a reverse set of experiments a triple Dpr11 mutant binds to DIP-a much weaker than wild-type Dpr10 – is it because of possible conformational changes? These residues are located in a flexible loop, could it be a source of this discrepancy? In general, I wonder how computational predictions of $\Delta\Delta G$ for reverse mutations would agree with the reverse set of experiments.
3. Negative constraints in the form of insertions and deletions (so called enabling or disabling loops) were studied previously, it might be a good idea to discuss different scenario when substitutions or insertions/deletions are used by evolution as negative constraints - does it depend on the type of interface and its properties?
4. "strengthens cognate binding to DIP-h by 0.3 kcal/mol" – such small changes are not significant.
5. The authors found that two thirds of interface utilize negative constraints – this fraction is similar to the fraction of destabilizing mutations in a protein or protein interface. Can the negative constraints reflect the fact that most interfacial mutations are destabilizing (unless there is some sort of enrichment), might be worth discussing it.

Reviewer #3 (Remarks to the Author):

The work by Sergeeva et al. addresses an important problem that arises continually in evolution: how do multi-protein families generate specific protein-protein interactions using the same overall tertiary structure. The authors identify 'negative constraints' that guide the evolution of specificity in the DIP/Dpr neuronal targeting proteins and test them in SPR experiments to validate the constraints they identified. The authors relied on protein structures and homology models to test the effect of mutations on the complex stability using a number of online servers for predicting mutational effects, eventually settling on FoldX, a physics-based force field approach, to make predictions. Then, they found by mutating a small number of important interface positions, they could change specificity of DIP/Dpr subfamilies. The study is sound and technically solid, but there are a few major concerns I have that have prevented me from giving the manuscript strong support. In addition, the study's breadth is narrow and it is unclear whether this would be of interest to the general scientific community.

Major concerns:

- 1) The study makes use of homology models and a host of online servers for the prediction of specificity-determining residues at the interface and their contributions to complex stability. There is no innovation in these techniques, so it is unclear whether this study reflects an advancement in our ability to uncover the determinants of specificity in a protein-protein interaction family.
- 2) The experimental work presented in this paper, while impressive, is restricted to two of the seven subfamilies. It would be useful to know the accuracy of the predictions made by FoldX would apply outside of these two subfamilies.

General notes:

It is unclear from the manuscript why the DIP/Dpr interaction has the same constraints on specificity that the Dscams/clustered protocadherins do? What is the biological consequence of interaction promiscuity?

"Negative constraints" is not well defined in the manuscript. My interpretation is that a negative constraint is a mutation or set of mutations that decrease the affinity of a subfamily to another (ancestral) family. Some definition should be included early on in the manuscript.

I don't quite understand the need for 42 distinct negative constraints in order for specificity. Take an example of 4 uniquely specific pairs. Given your calculation, there would be 16 possible pairs, with 4 that bind strongly, leading to your conclusion that there would be 12 sets of negative constraints. However, if one negative constraint differentiates the set into a group of two and two, and then two other negative constraints separate them into four groups of one, then only 3 sets of negative constraints are needed for the four subfamilies. Extending that to seven families, only 6 sets of negative constraints are needed. This can be thought of as a decision tree where each leaf should be isolated. At a minimum, 6 decisions are needed to isolate 7 subfamilies.

Are the homology models (or crystal structures) stable in simulations? How do the structures differ from the original models after equilibration/refinement in FoldX?

How well do changes in affinity by SPR correlate with FoldX calculations?

The authors make a comparison of the DIP/Dpr family to protocadherins, but do not cite their own work and others on understanding the specificity determinants of these interactions. The comparison to these approaches (and overall conclusions) and the one presented here would be greatly appreciated.

Specific notes:

Line 73. I'm a little confused about the description of identities: "average identity between individual DIPs and Dprs is about 30%." Is this a comparison of identity between a DIP and a Dpr? What is the conservation of DIP and Dpr orthologs?

Figure 1B. It looks as though the DIP-alpha and DIP-beta/DIP-gamma subfamilies should be swapped such that there are fewer Kd measurement lines that go through subfamilies. Is there a reason for the order as is?

Line 119. How many contacts are shared between subfamilies?

Line 123. "Throught" -> "throughout"

Line 125. A reference to figure should show up in this sentence instead of the next one.

Line 129. Stylistic - "of course" should not be placed between "cannot" and "be"

Line 151. In what ways were the number of sites expanded? Which ones were missed in your earlier studies and why based on your new analysis?

Line 192. What do these look like? It would be useful to look this over.

Line 204. Remove "It is clear from the table that"

Line 205. Where is the figure reference for "shown in green"?

Line 245. Instead of "qualitative" I think "biophysical" could be used to indicate that you are looking at biophysical properties of amino acid interactions. In fact, you use "physical" later in the manuscript.

Figure 3. From FoldX, it was found that position 10 would be more important to binding than position 29 and 31. I'm curious whether this is true experimentally. If you introduce mutations to position 10 and position 29 or 31 only, how does the affinity change for the parent species. Basically, how much can we trust the magnitude of the predictions?

Figure 4A. Why was position 6 identified as a negative constraint? It is cationic in both DIP-alpha and DIP-gamma. Is this negative constraint across all families or are they just between the subfamilies being tested?

General question for Figure 3 and 4. How did you select the positions to test? In some cases it looks like the highest negative constraint was not tested.

Line 306. Remove "It is evident from the figure that"

Line 330. Do you have an idea of the delta-delta G needed to ensure specificity based on this observation?

Figure 6. It is unclear to me how one would use this figure to determine the negative constraints between any two given subfamilies, i.e. what constitutes the 42 negative constraints that were described.

Line 385. This parenthesis is not closed.

Line 386. Change "constraints" to "negative constraints" for consistency throughout the manuscript.

Line 391. Can't several "tricks" be used in combinations, in some cases being important and in others, not?

Line 435. "dihedral" -> "dihedral"

Reviewer Response

Reviewer #1:

Sergeeva et al. report a detailed computational analysis of determinants of interaction specificity in Drosophila DIP and Dpr cell-cell recognition proteins. This system is intriguing because 7 subfamilies of proteins form preferred heterotypic interactions (DIP-to-Dpr protein interactions) without extensive cross-talk between subfamilies. The question addressed by the authors is how this intricate interaction specificity network is established at the structural level, which has been addressed before, but less comprehensively. Here the authors approach this question by building and analyzing homology models of numerous complexes and mutant complexes and predicting residues that can change binding specificity. They test selected predictions using binding experiments. An in-depth study of one pair of subfamilies includes experiments that confirm predictions of mutations that disrupt cognate complexes, as well as a nice example that shows how swapping appropriate residues into a weakly interacting complex can restore affinity. A fairly compelling picture emerges of how sets of distributed interactions are used to disfavor non-cognate DIP/Dpr complexes. The summary of the negative design strategies in Figure 5 provides a good overview. Overall the work is interesting, well done and makes a valuable contribution – by suggesting and illustrating general principles – that will shape thinking in the protein interaction field.

We thank the Reviewer for his/her overall positive assessment of our study.

There are some aspects of the paper that require clarification and the manuscript should be accompanied by a comprehensive report of the results of the analysis.

Three points related to Figure 1b. (1) Are these all fly proteins?

All the proteins in Fig. 1B are fruit fly proteins. We have modified the caption for Fig. 1B to remove the ambiguity: “(B) Affinity-based binding interactome of DIPs and Dprs of *Drosophila Melanogaster*.”

(2) I couldn't find an explanation of where the binding data in that figure panel come from. Are these values all from reference 11?

All the values in Fig. 1B are from reference 11. We now cite this reference in the caption.

The authors appear to assume that orthologs in the same subfamily conserve their interaction specificity, but they don't discuss why this is valid. This is a key point for the sequence-based analysis.

In response to this point, we have now added a sentence to the main text (lines 193-195): “The evolutionary filter assumes that orthologs in the same subfamily conserve their interaction specificity (see methods for data supporting the assumption).” We calculated pairwise sequence

identities in *Insecta* (per protein and per subgroup) and compared it to identities of interacting and non-interacting pairwise identities in *Drosophila Melanogaster* to justify this assumption (see methods, lines 567-577)

*As written, it is confusing whether the members of a subfamily, when used for a given purpose, include only fly proteins vs. all orthologous insect proteins. E.g. when residues were mutated “to all residues appearing at the same position in DIPs of other specificity subgroups,” did this consider only fly proteins? Lines 183 – 193 were particularly confusing in this regard. I am guessing there were two filters required to designate a residue as a negative constraint: (1) if analyzing fly subfamily *i*, the mutation must be predicted as disruptive when modeled in each fly protein member of subfamily *i* (giving $DDG > 0$ for at least one other, non-*i*, subfamily). (2) In a separate test, “we only defined a constraint as a position that would also be predicted for other *Insecta* species so as to focus on constraints that are evolutionarily conserved.” I gather this second prediction was not based on structural modeling, but rather on a qualitative comparison of biophysical properties, and applied to a much larger number of proteins (which makes sense; the residue equivalence categories used should be reported in the methods).*

Regardless of whether this is the correct interpretation of what was done, the authors should make this section easier to follow (e.g. by liberally adding the use of “fly” as a qualifier, in many places in the paper, and making clear there were 2 different tests at this stage). Then, on line 240 another criterion for a negative constraint is introduced: “In addition to this requirement, predictions are then filtered based on the requirement that $DDG > 0$ for all available blue subgroup crystal structures.” Along with the statement on line 220 that “Calculations are carried out on every complex for which a crystal structure is available.”

A general question: Were two different protocols used, one to study all fly DIP/Dpr subfamily pairs (using just one or two templates each) and a different one for complexes involving the blue/purple groups (using all available crystal structures and adding at least one new filter)? Or was there just one protocol that is described slightly differently in different places? If there were two different protocols, did the results differ? This was confusing.

We apologize for the confusion. The reviewer indeed understood correctly that there was one protocol with two filters: one based on energy (FoldX calculations of all available *Drosophila Melanogaster* structures), and the subsequent evolutionary filter based on analysis of properties of amino acids in multiple sequence alignments of DIP/Dpr subgroups in different species (*Insecta*). We have now clarified this in the main text (lines 195-197). We implemented the reviewer’s suggestion by adding qualifiers “energy filter” or “evolutionary filter” throughout the text.

The Reviewer’s comment about our qualitative sequence-based filter led us to develop an automated procedure which is now described in detail in Methods (lines 579-616). The new filter resulted in removal of about 45% of the FoldX predictions as opposed to about 30% in the original manuscript. However, none of the substantive conclusions of the paper were affected.

The paper should be accompanied by a list of the predicted negative constraints for all subfamilies, according to the described algorithm. Figure 6 highlights the positions that were found, but it would be good to include a comprehensive annotation (in the supplement) of which residues are predicted to destabilize which subfamily interactions. Or, even better, to make figure 5 for all subfamilies (given that the data are available).

We have now added a table listing negative constraints in every non-cognate DIP/Dpr pair (see Supplementary Table 4).

The paper says “Overall, we identified negative constraints for 42 combinations of non-interacting DIP and Dpr subgroups (see supplemental excel file for all FoldX data),” but I don’t see the list of negative constraints that passed the filters, nor was there an excel file among the materials I saw.

We apologize that we neglected to submit the supplementary excel file. It has now been submitted along with the revised version of the paper.

Line 133 states “We tested the ability of a number of sequence-based methods to identify specificity determinants in the seven specificity subgroups.” And line 153 says “Below we compare the predictions of these sequence-based methods to an energy-based analysis of DIP and Dpr structures.” To understand the result of that test and comparison, it should be easier for the reader to see which of these sequence-based predictions overlap with the structure-based predictions and which were experimentally verified in this work (or in other work). This could be added to Figure 6.

We thank the reviewer for this suggestion. The revised version of Fig. 6 now contains sequence-based predictions to facilitate comparison between different methods. The caption to Fig. 6 has also been appended with the following sentence: “Sequence-based predictions of specificity determining positions using GroupSim, SDPpred, SPEER, and Multi-Harmony methods (same as in Fig. 2A) are given below logos as specified in boxed inset.”

The paper states: “Similarly, we predict several residues in the hydrophobic core as potential negative constraints (12, 13, 16 in Dprs and 5, 13, 16 in DIPs), whereas sequence-based methods predict only one of these positions (Figure 2).” But these were not among the mutations that were experimentally confirmed to disrupt binding (I don’t think), which is worth mentioning here, since this may not be borne out.

We have added the following sentence (lines 378-379) to indicate the lack of supporting experimental data: “Of note, these positions were not among those we experimentally confirmed in this study.”

Experimental error estimates are lacking. I did not find descriptions of experimental error or fitting error, other than for 2 replicates of the AUC measurements. Were the SPR experiments performed more than once? (The checklist indicates replicates, but the paper does not, as far as

I saw). For K_D values, in Figure 3C and elsewhere, is the value in parentheses the error? E.g. 17.61(7) kcal/mol. What kind of error? In what units? How many replicates? Is this meant to indicate std. dev. of 0.07 kcal/mol (which seems way too low of an error for true replicates)? Related to this, many of the binding curves do not saturate. How was the 100% bound signal estimated, and is this well justified? The authors could report more conservative estimates of their measurement accuracies that would still substantiate their claims.

Most of our figures include a number in parenthesis next to each K_D measurement representing the error of the fit in the last significant figure in units of μM . We have clarified this point by including a statement in each legend for Fig. 3 and Supplementary Figs. 1, 3 and 4 stating that “for each K_D , the number in parenthesis represents the fitting error in the last significant figure, in μM ”. Errors are reported for K_D measurements only. In addition, we have modified the errors in Supplementary Fig. 4 from two to one significant figure to more realistically reflect inaccuracies in our measurements. We have also incorporated errors of the fit for K_D s shown in Figs. 3A, 4, and Supplementary Figs. 2 and 3D.

We have added a description of the experimental details that explains the use of two technical replicates in the Methods section of our SPR experiments: “Within each experiment, there two technical replicates starting with a single concentration series, where samples are tested in order of increasing concentration, followed by a repeat of the same concentration series, performed again from low to high concentration”. An explanation was also included in the SPR Methods section that describes the fitting of duplicate responses to calculate a single K_D and the fitting error: “The duplicate binding responses were fit globally, using a 1:1 interaction model and a single K_D was calculated as the analyte concentration that would yield $0.5 R_{\text{max}}^{46}$ and a fitting error”. We have also included data for experimental errors determined from independent experiments for Dpr10/DIP- α , Dpr6/DIP- α , and Dpr11/DIP- γ . “For several wild type interactions discussed in this manuscript, Dpr10/DIP- α , Dpr6/DIP- α , and Dpr11/DIP- γ , we have also determined K_D s from independent experiments. For the Dpr10/DIP- α binding pair, the K_D from six independent experiments is $1.5 \pm 0.1 \mu\text{M}$, and similarly for the Dpr6/DIP- α , the K_D from six independent experiments is $2.0 \pm 0.2 \mu\text{M}$. The K_D from five independent experiments for Dpr11/DIP- γ is $7.9 \pm 0.9 \mu\text{M}$.”

We now report average values next to the FoldX data in Figs. 3A, 4, and Supplementary Figs. 2 and 3D and throughout the main text of the manuscript. However, below individual SPR binding curves we report the K_D s obtained from that experiment and indicate fitting errors as well (Figures 3C, 3D, and Supplementary Figs. 1, 3, 4).

The Methods section now contains a discussion of the extraction of K_D values when binding is weak and only one of the binding curves reaches saturation (lines 698-700).

Minor points:

Several different and contradictory values for the dissociation constant of DIP-g/Dpr11 are reported (7.1, 7.9, 8.5) in the text and figures.

We thank the reviewer for pointing out this inconsistency, which arose due to reporting values from either a specific SPR experiment or the average K_D (7.9 ± 0.9 mM) of the five independent SPR experiments using the same buffer and sample preparation conditions. We now report average values next to the FoldX data in Figs. 3A, 4, and Supplementary Figs. 2 and 3D and throughout the main text of the manuscript. However, below individual SPR binding curves we report the K_D s obtained from that experiment and indicate fitting errors as well (Figures 3C, 3D, and Supplementary Figs. 1, 3, 4).

It would be helpful to clarify why there is a stochastic element of the FoldX modeling procedure. E.g. to explain why 10 runs gave different values. Evaluating mutational energies by comparing mutant and WT structures with the same rotamers in each structure is an unusual procedure (though I can imagine it may cancel errors). Can the authors explain/justify this? And explain how the structure that was used was selected? I.e., from the methods: "This requires initial sampling of different rotamers for a mutated residue and subsequent synchronous movement of neighboring residues in a WT complex and a MT complex to find the optimal position for the mutated residue." But what is "optimal" in this context, given there are two different structures to consider?

We apologize for the confusion and, indeed, our description of the FoldX procedure was too vague. The various FoldX routines are now described in greater detail in Methods. With regard to the specific question, since the FoldX procedure begins with a random choice of a rotamer and then the minimization procedure cycles through rotamer libraries for the mutated residue and its neighbors, it is not guaranteed that each run yields the same results. This is why we calculate an average of 10 runs. The WT and MT runs are required to involve the same side chains but the use of the word synchronous was misleading. We hope that our explanation in Methods is now clear (lines 502-522).

I didn't understand this expression: $33[3(10+19) + 1(10+17) + 5(8+16) + 1(10+20) + 1(9+18) + 1(10+20) + 1(9+16)] = 11418$ mutations. First, is the "3" in "3(10+19)" meant to be "2" (for "two conformations of Dpr10/DIP")? It would be good to clarify the source of the terms in this expression.*

Since it is not really important for the content of the paper, we have removed the expression from the text and just state how many mutations were carried out.

Next, if the numbers like 10, 17, 8, 16, etc. are the numbers of proteins in non-i subgroups, this seems like an overestimate of the number of mutations to be made, given that many proteins share the same interface residue at a given site.

The reviewer is correct and indeed we carried out calculations unnecessarily based on the scripts that were written originally. The text has now been changed to give the number of mutations that were actually required.

Reviewer #2:

This is a very nice paper that explores the basis of binding specificity using an example of a physical interaction between proteins from DIP and Dpr families. The authors performed in silico mutational screening and found that some amino acid substitutions serve as negative constraints and prevent certain undesired interactions between members of these families. Moreover, the authors went further and performed experimental binding affinity measurements to confirm their computational findings. I have several suggestions which can potentially improve the paper:

1. It might be worth discussing that stabilizing mutations can also represent negative constraints since the studied interaction is rather weak and transient.

This issue is discussed in part in the section titled “Balancing affinity and specificity” (lines 380-394). We have also added a discussion of this general issue (lines 444-450).

2. In a reverse set of experiments a triple Dpr11 mutant binds to DIP- α much weaker than wild-type Dpr10 – is it because of possible conformational changes? These residues are located in a flexible loop, could it be a source of this discrepancy? In general, I wonder how computational predictions of $\Delta\Delta G$ for reverse mutations would agree with the reverse set of experiments.

In response to the reviewer’s question, we ran additional FoldX calculations on a hypothetical Dpr11/DIP- α complex that doesn’t form to compare with the reverse set of mutations probed experimentally in Figure 3D. Two of the mutations qualitatively agree with experiment ($\Delta\Delta G(\text{FoldX}, \text{G10L}) = -0.9 \text{ kcal/mol}$ vs. $\Delta\Delta G(\text{SPR}, \text{G10L}) < -2.0 \text{ kcal/mol}$; $\Delta\Delta G(\text{FoldX}, \text{E29Q}) = 0.1 \text{ kcal/mol}$ vs. $\Delta\Delta G(\text{SPR}, \text{E29Q}) \sim 0 \text{ kcal/mol}$), and one mutation does not agree ($\Delta\Delta G(\text{FoldX}, \text{K31V}) = 1.2 \text{ kcal/mol}$ vs. $\Delta\Delta G(\text{SPR}, \text{K31V}) \sim -0.3 \text{ kcal/mol}$). Notably, the FoldX success rate for prediction of stabilizing mutations is much lower compared to destabilizing mutations (O. Buß et al., Comput. Struct. Biotechnol. J., 2018, 16, 25-33). This is the reason why our computational strategy relied on mutating cognate complexes that do form to find mutations that results in destabilizing effects, rather than predicting stabilizing mutations in non-cognate hypothetical complexes that do not form. In addition, there are likely to be factors that contribute to affinity that we have missed, including residues not in the interface or, as pointed out by the reviewer, effects associated with flexible loops.

3. Negative constraints in the form of insertions and deletions (so called enabling or disabling loops) were studied previously, it might be a good idea to discuss different scenario when substitutions or insertions/deletions are used by evolution as negative constraints - does it depend on the type of interface and its properties?

We have now pointed out in the Discussion that negative constraints could potentially manifest via loop insertions/deletions, though this strategy is not used by DIPs/Dprs (line 458-460).

4. *“strengthens cognate binding to DIP-h by 0.3 kcal/mol” – such small changes are not significant.*

We have estimated that the standard deviation of the $\Delta\Delta G$ value based on experimentally measured affinities is less than 0.1 kcal/mol (this information has been added to Methods, lines 706-708). Therefore, the above experimental value of 0.3 kcal/mol is likely to be significant. Nevertheless, we have added a phrase acknowledging that this is a small effect.

5. *The authors found that two thirds of interface utilize negative constraints – this fraction is similar to the fraction of destabilizing mutations in a protein or protein interface. Can the negative constraints reflect the fact that most interfacial mutations are destabilizing (unless there is some sort of enrichment), might be worth discussing it.*

We think the cases discussed here are different since we are looking at a large family where much of the interface is conserved so the connection to the general finding mentioned by the referee is admittedly unclear to us.

Reviewer #3 (Remarks to the Author):

The work by Sergeeva et al. addresses an important problem that arises continually in evolution: how do multi-protein families generate specific protein-protein interactions using the same overall tertiary structure. The authors identify 'negative constraints' that guide the evolution of specificity in the DIP/Dpr neuronal targeting proteins and test them in SPR experiments to validate the constraints they identified. The authors relied on protein structures and homology models to test the effect of mutations on the complex stability using a number of online servers for predicting mutational effects, eventually settling on FoldX, a physics-based force field approach, to make predictions. Then, they found by mutating a small number of important interface positions, they could change specificity of DIP/Dpr subfamilies. The study is sound and technically solid, but there are a few major concerns I have that have prevented me from giving the manuscript strong support. In addition, the study's breadth is narrow and it is unclear whether this would be of interest to the general scientific community.

Major concerns:

1) *The study makes use of homology models and a host of online servers for the prediction of specificity-determining residues at the interface and their contributions to complex stability. There is no innovation in these techniques, so it is unclear whether this study reflects an advancement in our ability to uncover the determinants of specificity in a protein-protein interaction family.*

The novel application of existing methods has the potential to uncover new principles and insights even if no new methodology is reported. In this case the innovation is not in the techniques that are used, but rather in the approach involving the large-scale use of homology models to study the design of protein interaction specificity. To our knowledge, no study of this kind, and certainly not of this scale, has previously been reported. Moreover, the problem of interfamily specificity is of broad relevance not only to adhesion proteins, but to many cytoplasmic proteins as well. Thus, despite the focus on a single family, the insights we have gained and the method used to gain them should be of quite general relevance to the study of other protein families.

2) The experimental work presented in this paper, while impressive, is restricted to two of the seven subfamilies. It would be useful to know the accuracy of the predictions made by FoldX would apply outside of these two subfamilies.

We previously published experimental results regarding specificity determinants for green and blue subfamilies (Cosmanescu et al., Neuron, 2018, 100, 1385-1400). Specifically, we tried to switch the specificity of Dpr4 and Dpr6 by mutating Dpr6 to make it bind DIP- η instead of DIP- α and making Dpr4 bind to DIP- α instead of DIP- η . We have now added Supplementary Table 3 which compares FoldX calculations with these data and discuss the table on lines 323-336. In short, the calculations are in very good qualitative agreement with the experiments.

General notes:

It is unclear from the manuscript why the DIP/Dpr interaction has the same constraints on specificity that the Dscams/clustered protocadherins do? What is the biological consequence of interaction promiscuity?

In fact, Dscams/cPcdhs have far more restrictive negative constraints than DIPs and Dprs. In the former case, promiscuity would disrupt their function in the precise barcoding of neurons. This is why these proteins are *strictly* homophilic. The specificity constraints on DIPs/Dprs are clearly less strict but still strict enough to enable inter-neuronal targeting. Further *in vivo* experiments will be required to determine in detail the role of specificity and promiscuity in targeting and, as we now point out, our results are enabling studies of this type (with Larry Zipursky) – another reason why we believe that our paper will be of broad interest. We have now added a paragraph to the discussion where we consider these issues (lines 473-482).

"Negative constraints" is not well defined in the manuscript. My interpretation is that a negative constraint is a mutation or set of mutations that decrease the affinity of a subfamily to another (ancestral) family. Some definition should be included early on in the manuscript.

We apologize for our lack of precision. The reviewer is correct and we have now added a clear definition (lines 80-84).

I don't quite understand the need for 42 distinct negative constraints in order for specificity. Take an example of 4 uniquely specific pairs. Given your calculation, there would be 16 possible pairs, with 4 that bind strongly, leading to your conclusion that there would be 12 sets of negative constraints. However, if one negative constraint differentiates the set into a group of two and two, and then two other negative constraints separate them into four groups of one, then only 3 sets of negative constraints are needed for the four subfamilies. Extending that to seven families, only 6 sets of negative constraints are needed. This can be thought of as a decision tree where each leaf should be isolated. At a minimum, 6 decisions are needed to isolate 7 subfamilies.

This is a fair point in that in principle things could work in this relatively simple way but this seems not to be the case. We now provide a Supplemental Table 3 lists constraints for every non-cognate DIP/Dpr pair. The table shows that there are no identical sets of constraints for any DIP/Dpr pair. We now point this out in the discussion (lines 427-428).

Are the homology models (or crystal structures) stable in simulations? How do the structures differ from the original models after equilibration/refinement in FoldX?

FoldX is a rigid backbone method, therefore, the backbone of the structure is unchanged. Only side-chain positions are minimized so no true simulation is involved.

How well do changes in affinity by SPR correlate with FoldX calculations?

Based on twenty-two mutations from our dataset that were assessed by SPR (see Supplementary Table 2), the Pearson correlation coefficient (PCC) between FoldX and SPR is 0.57. We now note this in the manuscript (line 222).

The authors make a comparison of the DIP/Dpr family to protocadherins, but do not cite their own work and others on understanding the specificity determinants of these interactions. The comparison to these approaches (and overall conclusions) and the one presented here would be greatly appreciated.

A discussion of these issues has now been added to the manuscript. We appreciate the suggestion and believe that this will add to the interest in our work. On the other hand, specificity determinants for Dscams and Protocadherins have not yet been described in detail.

Specific notes:

Line 73. I'm a little confused about the description of identities: "average identity between individual DIPs and Dprs is about 30%." Is this a comparison of identity between a DIP and a Dpr? What is the conservation of DIP and Dpr orthologs?

The reported numbers are for DIPs and Dprs from one species, *Drosophila Melanogaster*. We changed the above sentence to avoid confusing these values with identity between DIP and Dpr

orthologs: “The Ig1 domains of the *Drosophila Melanogaster* DIPs and Dprs have intra-family pairwise sequence identities greater than about 50% and 40%, respectively, while the average identity between individual DIPs and Dprs is about 30%.”

We have evaluated pairwise identities for interfacial residues between orthologs of DIP and a Dpr in *Insecta* species to justify the assumption that specificity is conserved throughout orthologs (see methods, lines 567-577).

Figure 1B. It looks as though the DIP-alpha and DIP-beta/DIP-gamma subfamilies should be swapped such that there are fewer Kd measurement lines that go through subfamilies. Is there a reason for the order as is?

There was no specific reason for the order. We have now changed the order as suggested by the reviewer. Thanks!

Line 119. How many contacts are shared between subfamilies?

Nearly all the interfacial contacts are expected to be shared between subfamilies due to high similarity of Ig1-Ig1 interfaces (RMSD <1Å).

Line 125. A reference to figure should show up in this sentence instead of the next one. We have moved the reference to Fig. 2A as the reviewer suggests.

Line 151. In what ways were the number of sites expanded? Which ones were missed in your earlier studies and why based on your new analysis?

15 sites were missed in our earlier studies. The previous study relied on finding specificity positions by manual inspection of sequence alignments. Sometimes a specificity site is not apparent by eye if positions are not fully conserved within a subfamily or if specificity sites are specific to only one subfamily while being conserved throughout others. For example, position 10 in Dprs was not picked up in the previous study because this position was not conserved within four subfamilies and because identical amino acids were present in five subfamilies.

Line 192. What do these look like? It would be useful to look this over.

The revised submission includes a supplementary excel file with FoldX-based predictions of negative constraints.

Line 204. Remove "It is clear from the table that"

Done

Line 205. Where is the figure reference for "shown in green"?

We have added a reference to Supplementary Table 2 in the revised sentence: “MutaBind is the best performer but it fails to identify stabilizing mutations (shown in green, see Supplementary Table 2)...”

Line 245. Instead of "qualitative" I think "biophysical" could be used to indicate that you are looking at biophysical properties of amino acid interactions. In fact, you use "physical" later in the manuscript.

We agree with the reviewer that biophysical is the better choice of words.

Figure 3. From FoldX, it was found that position 10 would be more important to binding than position 29 and 31. I'm curious whether this is true experimentally. If you introduce mutations to position 10 and position 29 or 31 only, how does the affinity change for the parent species. Basically, how much can we trust the magnitude of the predictions?

These experiments were not carried out as our main goal was to determine how many constraints were needed to kill binding. We admit to being curious about the question raised by the referee as well, but feel that the table we provided (Supplementary Table 2) gives a general picture of FoldX reliability.

Figure 4A. Why was position 6 identified as a negative constraint? It is cationic in both DIP-alpha and DIP-gamma. Is this negative constraint across all families or are they just between the subfamilies being tested?

FoldX identifies Arg to Lys mutation as destabilizing in a number of DIP/Dpr families possibly due to the longer Arg side chain (in the above case allowing Arg to form a tighter salt bridge with Glu on Dpr10) but also due to the more delocalized charge in Arg which results in a smaller desolvation penalty of burying an Arg than a Lys residue in an interface (see e.g. Rohs et al., Nature, 2009, 461, 1248–1253 and Hwang et al. Protein Sci, 2016, 25, 159-165).

General question for Figure 3 and 4. How did you select the positions to test? In some cases, it looks like the highest negative constraint was not tested.

Can you answer briefly. Were there experimental reasons for the choices.

In Fig. 3, we originally identified four constraints (at positions 10,15,29,31) and we chose to test only three because position 15 was previously tested by Ozkan and co-workers (Cheng et al., Elife, 2019, 8, e41028). In any case, position 15 did not pass our revised evolutionary filter.

In Fig. 4, we tested a subset of all the predictions by mostly focusing on mutations that uncouple charge-charge pairs, in part because previous work has indicated that these do not play a role in specificity (Carillo et al, Cell, 2015, 163, 1770-1782). This surprised us as we have found that electrostatics plays a crucial role for specificity in other protein families (Harrison et al, Nat. Struct. Mol. Biol, 2012, 19, 906-915; Harrison et al., PNAS, 2016, 113, 7160-7165).

Line 306. Remove "It is evident from the figure that"

Done

Line 330. Do you have an idea of the delta-delta G needed to ensure specificity based on this observation?

We believe we address this question in the discussion where we wrote:

“As can be seen in Fig. 1B, intra-subgroup K_{DS} are generally in the range of 10 μM (although some are as weak as 40 μM) while inter-subgroup K_{DS} (with a few exceptions) are undetectable, implying a $K_D > 500 \mu\text{M}$. This suggests that the generation of a new DIP/Dpr specificity requires a change in binding free energy of $\sim 2.5 \text{ kcal mol}^{-1}$, a value that is difficult (but not impossible) to reach with one or two mutations”.

Figure 6. It is unclear to me how one would use this figure to determine the negative constraints between any two given subfamilies, i.e. what constitutes the 42 negative constraints that were described.

This question is now addressed by providing a supplemental excel file mentioned above.

Line 391. Can't several "tricks" be used in combinations, in some cases being important and in others, not?

Yes, that is what we predict. To make sure this conclusion is more transparent to the readers, we modified the text of the manuscript to specifically say it (line 426-429): “Of course, the same “trick” cannot be reused for 42 different subgroup pairs. In fact, every set of negative constraints on the DIP and Dpr side of the 42 non-cognate pairs is unique in the sense that the same set is not reused (see Supplementary Table 4). This is why as discussed above, most of the interface needs to be exploited.”

We also thank the reviewer for pointing out typos, inconsistent use of terms, and stylistic mistakes in our manuscript:

Line 123. "Thought" -> "throughout"

The typo has been fixed in the revised version of the manuscript.

Line 129. Stylistic - "of course" should not be placed between "cannot" and "be"

Corrected.

Line 385. This parenthesis is not closed.

Corrected.

Line 386. Change "constraints" to "negative constraints" for consistency throughout the manuscript.

Done

Line 435. "dihedral" -> "dihedral"
Corrected.

Reviewers' comments:

Reviewer #1 (Remarks to the Author):

My comments from the first round of review have mostly been addressed.

However, if I understand the newly included explanation of the experimental error analysis correctly, then the reported K_d values are misleading. Very small errors are reported that are the fitting error from two technical replicates from re-measuring *the same samples* (in reverse order). This does not provide an estimate of measurement error. I.e. this error estimate does not account for any pipetting error in constructing the concentration series, error in sample stock concentration determination, or in batch-to-batch variation. It doesn't control for grabbing the wrong tube. It seems that the authors are drawing quantitative conclusions from measurements that were performed once, controlling only for things like instrument drift.

Also related to the experimental binding curves, this statement (new in this version) is not sufficiently specific, and it is not clear if what was done in curve fitting is justified: "In cases where the highest analyte concentration is only around the K_D , we use the R_{max} value for a saturating binding isotherm shown on the same plot, as a global R_{max} , to fit the binding isotherms for weaker interactions and calculate a K_D ." By "on the same plot" do the authors mean that, e.g., for Figure 3C the saturated value for wild-type Dpr10 was used for all Dpr10 mutants? Is there evidence this is valid? Was the same thing done to compare curves with different mutant proteins on the surface (e.g. Supp. Figure 1)? This requires an assumption of similar loading from experiment to experiment because the same surface isn't being re-used. This requires better justification.

Given the way that the K_d values were generated (no real replicates), the errors are certainly much larger than what is reported. A few values could be grossly wrong. There is a serious risk to this subfield that these values will be accepted as "real" and propagated in the literature.

There are two possible routes forward:

- (1) Much preferred: Perform replicate experiments (ideally at least 3 total replicates from different days/pipetting series) and report measurement errors that allow an assessment of what comparisons are justified
- (2) If replicates can't be performed for some reason: Because the overall conclusions of the paper do not depend critically on any single measurement, and there are trends in the data that suggest that many of the authors' predictions/residue assignment are valid, the authors could make it very clear that they are basing their conclusions on approximate/estimated K_d values from single measurements that have not been reproduced. In this case, a clear caveat should appear in the main text, and all tables/figure legends should refer to the values as "estimated K_d values" and use some symbol like K_d^* or K_{d_est} . There is a concern that the appropriate caveats associated with these values will be lost in future work, if they are cited, but at least an attentive reader who checks the original source will be alerted to the uncertainty.

In general, there is now greater clarity with respect to what was done in the computational work, but there are still some places where the wording is insufficiently precise. I recommend another round of careful editing (also to fix typos).

Two examples that were added in this version are:

- (1) the description of the 3 cases for the evolution filter on p. 29 and following is not sufficiently precise that one could write code to reproduce the values (it is ambiguous what "we summed the instances where amino acids at a position of subfamily "i" (aai) were different in biophysical property from aaj" means. Though the general gist of the approach is clear, it would be better to

give an equation, since one cannot understand from what is written what set of "instances" was summed over. This doesn't affect the main conclusions.

(2) this sentence was very difficult to parse:

To justify this, we calculated the pairwise sequence identities within Insecta orthologs for each DIP and Dpr, and the intra-subgroup pairwise sequence identity for each DIP and Dpr Insecta subgroup (~78%) using multiple sequence alignments of interfacial residues." How are "the pairwise sequence identities within Insecta orthologs for each DIP and Dpr" (no value given) different from "the intra-subgroup pairwise sequence identity for each DIP and Dpr Insecta subgroup (~78%)."
This doesn't affect the conclusions but is confusing.

Reviewer #3 (Remarks to the Author):

I appreciate the author's effort in addressing the comments from all reviewers. I feel the manuscript addresses these concerns. In particular, I will address a few points that I made and how I feel the authors appropriately modified the manuscript:

1) It is more clear to me that the biggest innovation of the article is the use of homology models in computationally determining protein interaction specificity. I appreciate the difficulty in producing accurate homology models and then testing these for specificity determinants. The only questions that remains for me is whether FoldX calculations for interactions from homology models are worse than for FoldX calculations for actual structures. Answering this would address whether homology models are sufficient for determining protein interaction specificity.

2) I appreciate the inclusion of Supplementary Table 3, which expands the analysis of specificity to include those that were previously studied.

3) I now appreciate that the DIP/Dpr interactions are less specific than Dscams or protocadherins and look forward to the groups future work on understanding how the level of specificity affects phenotype. I think inclusion of a deeper comparison to specificity of Dscams and clustered protocadherins could help put this work in better context for the reader. The authors mention (lines 467-472) that specificity is spread across multiple domains in these families, which results in greater specificity. This result is addressed specifically in several publications that are not cited.

4) On the issue of the number of negative constraints needed, I appreciate the inclusion of Supplementary Table 4, which describes the negative constraints they found. I also now understand that no two sets of negative constraints are the same between any two non-cognate pairs.

Finally, here is a minor change to the corrections:

Line 52 (and others): *Drosophila melanogaster* ('m' should not be capitalized)

Reviewer Response

Reviewer #1:

My comments from the first round of review have mostly been addressed.

*However, if I understand the newly included explanation of the experimental error analysis correctly, then the reported K_d values are misleading. Very small errors are reported that are the fitting error from two technical replicates from re-measuring *the same samples* (in reverse order). This does not provide an estimate of measurement error. I.e. this error estimate does not account for any pipetting error in constructing the concentration series, error in sample stock concentration determination, or in batch-to-batch variation. It doesn't control for grabbing the wrong tube. It seems that the authors are drawing quantitative conclusions from measurements that were performed once, controlling only for things like instrument drift.*

Given the way that the K_d values were generated (no real replicates), the errors are certainly much larger than what is reported. A few values could be grossly wrong. There is a serious risk to this subfield that these values will be accepted as "real" and propagated in the literature.

There are two possible routes forward:

(1) Much preferred: Perform replicate experiments (ideally at least 3 total replicates from different days/pipetting series) and report measurement errors that allow an assessment of what comparisons are justified

(2) If replicates can't be performed for some reason: Because the overall conclusions of the paper do not depend critically on any single measurement, and there are trends in the data that suggest that many of the authors' predictions/residue assignment are valid, the authors could make it very clear that they are basing their conclusions on approximate/estimated K_d values from single measurements that have not been reproduced. In this case, a clear caveat should appear in the main text, and all tables/figure legends should refer to the values as "estimated K_d values" and use some symbol like K_d^ or K_{d_est} . There is a concern that the appropriate caveats associated with these values will be lost in future work, if they are cited, but at least an attentive reader who checks the original source will be alerted to the uncertainty.*

Since option (1) would require months of additional work likely resulting in minimal changes, if any, to the manuscript, we have preferred option (2). Specifically, we have provided the reader with all the necessary information to evaluate the accuracy of the K_D s we report.

As described in the original Methods section, and now in Results (lines 268-275), the experiments on three wild-type protein complexes were done with six replicated experiments in two cases and five replicated experiments in one case. Notably, these experiments were performed over the course of two years, over individually immobilized chips, different buffer preparations, separate dilution series and with distinct protein preparations prepared by different scientists. The experimental error ranged between 10% and 15% for these replicated

experiments and we expect that a similar level of experimental error applies to all the other SPR measurements on mutant proteins. In addition to the text, we have made it very clear in the figure captions (Figures 3, 4, Supplemental Figures 1, 3 and 4) that the K_D values we report reflect this error.

Also related to the experimental binding curves, this statement (new in this version) is not sufficiently specific, and it is not clear if what was done in curve fitting is justified:

"In cases where the highest analyte concentration is only around the K_D , we use the R_{max} value for a saturating binding isotherm shown on the same plot, as a global R_{max} , to fit the binding isotherms for weaker interactions and calculate a K_D ." By "on the same plot" do the authors mean that, e.g., for Figure 3C the saturated value for wild-type Dpr10 was used for all Dpr10 mutants? Is there evidence this is valid? Was the same thing done to compare curves with different mutant proteins on the surface (e.g. Supp. Figure 1)? This requires an assumption of similar loading from experiment to experiment because the same surface isn't being re-used. This requires better justification.

We understand the reviewer's concern and we apologize for not being clearer in that we only used R_{max} values for analytes tested over the same surface in the same experiment. As now clarified in the Methods section (lines 732-737), this analysis approach is standardly used for weak interactions. Although the responses for some of the mutants do not saturate, experiments that provide highly reproducible binding profiles and use a 1:1 binding model, can extract accurate K_D s from equilibrium binding responses even with binding occupancy as low as 44% as previously discussed by Rich and Myszka (Rich, R., & Myszka, D., 2009, Extracting affinity constants from biosensor binding responses. In M. Cooper (Ed.), Label-Free Biosensors: Techniques and Applications (pp. 48-84). Cambridge: Cambridge University Press. doi:10.1017/CBO9780511626531.005).

In general, there is now greater clarity with respect to what was done in the computational work, but there are still some places where the wording is insufficiently precise. I recommend another round of careful editing (also to fix typos).

Two examples that were added in this version are:

(1) the description of the 3 cases for the evolution filter on p. 29 and following is not sufficiently precise that one could write code to reproduce the values (it is ambiguous what "we summed the instances where amino acids at a position of subfamily "i" (aai) were different in biophysical property from aaj" means. Though the general gist of the approach is clear, it would be better to give an equation, since one cannot understand from what is written what set of "instances" was summed over. This doesn't affect the main conclusions.

We have now provided a more detailed protocol including relevant equations (see lines 628-637,645-654).

(2) this sentence was very difficult to parse:

To justify this, we calculated the pairwise sequence identities within Insecta orthologs for each DIP and Dpr, and the intra-subgroup pairwise sequence identity for each DIP and Dpr Insecta subgroup (~78%) using multiple sequence alignments of interfacial residues." How are "the pairwise sequence identities within Insecta orthologs for each DIP and Dpr" (no value given) different from "the intra-subgroup pairwise sequence identity for each DIP and Dpr Insecta subgroup (~78%)." This doesn't affect the conclusions but is confusing.

We apologize for the confusion. We have done our best to justify our contention that orthologs conserve their interaction specificity with the following modified sentences: "To justify this, we calculated the average pairwise sequence identities within *Insecta* orthologs for interfacial residues of each DIP and Dpr. The average pairwise sequence identity for interfacial residues of DIP (86%) and Dpr (83%) orthologs in *Insecta* is 84%. This value is similar to the average intra-subgroup pairwise sequence identity for each DIP and Dpr *Insecta* subgroup (78%) and the intra-subgroup pairwise sequence identities of interfacial residues in *Drosophila melanogaster* (~80%). These values are also consistently higher than inter-subgroup pairwise sequence identities in *Drosophila melanogaster* or *Insecta* (~50%)." (lines 592-598).

Reviewer #3:

I appreciate the author's effort in addressing the comments from all reviewers. I feel the manuscript addresses these concerns. In particular, I will address a few points that I made and how I feel the authors appropriately modified the manuscript:

1) It is more clear to me that the biggest innovation of the article is the use of homology models in computationally determining protein interaction specificity. I appreciate the difficulty in producing accurate homology models and then testing these for specificity determinants. The only questions that remains for me is whether FoldX calculations for interactions from homology models are worse than for FoldX calculations for actual structures. Answering this would address whether homology models are sufficient for determining protein interaction specificity.

Our study only uses high-quality homology models based on high sequence identities and the fact that all DIP/Dpr crystal structures superimpose to within 1Å. Thus, we expect the FoldX calculations on homology models to perform similarly to calculations on crystal structures. To justify this point, we carried out FoldX calculations on homology models and compared the results to those obtained from crystal structures of Dpr11/DIP- γ , Dpr6/DIP- α , and Dpr10/DIP- α . The Pearson correlation coefficient (and root mean square error) with experimental values was 0.56 (1.65) for the crystal structures and 0.53 (1.10) for the homology models based on 25 experimental measurements of binding energy differences. On this basis, we now point out in the Methods section that our homology models give comparable FoldX results to those obtained from crystal structures (lines 535-542).

2) *I appreciate the inclusion of Supplementary Table 3, which expands the analysis of specificity to include those that were previously studied.*

3) *I now appreciate that the DIP/Dpr interactions are less specific than Dscams or protocadherins and look forward to the groups future work on understanding how the level of specificity affects phenotype. I think inclusion of a deeper comparison to specificity of Dscams and clustered protocadherins could help put this work in better context for the reader. The authors mention (lines 467-472) that specificity is spread across multiple domains in these families, which results in greater specificity. This result is addressed specifically in several publications that are not cited.*

We have corrected our oversight and now provided a complete set of references. We added an additional sentence how multi-domain interfaces can be exploited (see lines 481-483). As the reviewer recognizes, a more extensive analysis must await a detailed study of negative constraints in Pcdhs and DSCAMs.

4) *On the issue of the number of negative constraints needed, I appreciate the inclusion of Supplementary Table 4, which describes the negative constraints they found. I also now understand that no two sets of negative constraints are the same between any two non-cognate pairs.*

Finally, here is a minor change to the corrections:

Line 52 (and others): *Drosophila melanogaster* ('m' should not be capitalized)

Done